# Guaranteeing Robustness Against Real-World Perturbations In Time Series Classification Using Conformalized Randomized Smoothing

Nicola Franco[1]          Jakob Spiegelberg[2]          Jeanette Miriam Lorenz[1]          Stephan Günnemann[3]

[1]Fraunhofer Institute for Cognitive Systems IKS, Munich, Germany
[2]Volkswagen Group Innovation, Volkswagen AG, Wolfsburg, Germany
[3]School of Computation, Information & Technology, Technical Univ. of Munich, Germany

## Abstract

Certifying the robustness of machine learning models against domain shifts and input space perturbations is crucial for many applications, where high risk decisions are based on the model's predictions. Techniques such as randomized smoothing have partially addressed this issues with a focus on adversarial attacks in the past. In this paper, we generalize randomized smoothing to arbitrary transformations and extend it to conformal prediction. The proposed ansatz is demonstrated on a time series classifier connected to an automotive use case. We meticulously assess the robustness of smooth classifiers in environments subjected to various degrees and types of time series native perturbations and compare it against standard conformal predictors. The proposed method consistently offers superior resistance to perturbations, maintaining high classification accuracy and reliability. Additionally, we are able to bound the performance on new domains via calibrating generalisation with configuration shifts in the training data. In combination, conformalized randomized smoothing may offer a model agnostic approach to construct robust classifiers tailored to perturbations in their respective applications - a crucial capability for AI assurance argumentation.

## 1 INTRODUCTION

Deep neural networks have shown their remarkable ability to learn intricate patterns from vast amounts of data, marking them as a preferred choice for complex challenges [LeCun et al., 2015, Han et al., 2022, Wen et al., 2022]. In order to be able to employ these models in applications where high-stake decisions are based on their outputs, such as in all safety-critical systems, it is imperative not only to ensure their accuracy but also to understand and quantify the confidence attached to their predictions and the robustness of the model to input space perturbations and domain shifts. Addressing the former, the Conformal Prediction (CP) [Vovk et al., 1999, 2005, Shafer and Vovk, 2008] framework has emerged as a tool to construct set classifiers with guaranteed confidence that can be adjusted by the user.

The importance of *domain generalization* becomes especially crucial when the costs of mispredictions are high, as in medical diagnosis or autonomous driving [Zhou et al., 2022, Wang et al., 2022]. Notably, research by Park et al. [2019, 2020, 2022a,b] demonstrated how CP can be utilized to bound performance on unseen domains, if there is an adequate amount of training domains for calibration during the training process. In addition, Randomized Smoothing (RS) [Cohen et al., 2019, Salman et al., 2019] has emerged as method to robustify machine learning models against worst case, i.e., adversarial, attacks. While undeniably successful as defense mechanism, RS is not equally useful in an assurance argumentation, where it is necessary to demonstrate robustness against disturbances inherent to the specific data domain being targeted.

Often, real life perturbations come with large norms which renders certification with respect to some $\ell_p$-norm - as typically done in RS - prohibitively conservative, as huge volumes would need to be certified. Consider the example in Figure 1a, where a time series signal is characterized by a few peaks. When a time warping augmentation is applied, it may result in a large magnitude for a conventional $\ell_p$-norm. However, this transformation would remain concentrated around the original peak's location. As a result, in order to establish certified robustness, we do not need to consider other transformations with equal norm but amplitudes at a different position of the signal. These considerations motivate the new technique developed in this paper.

**Contribution** In this paper, we generalize RS to arbitrary perturbations following an automotive use case build around a binary classification of time series input. The input time

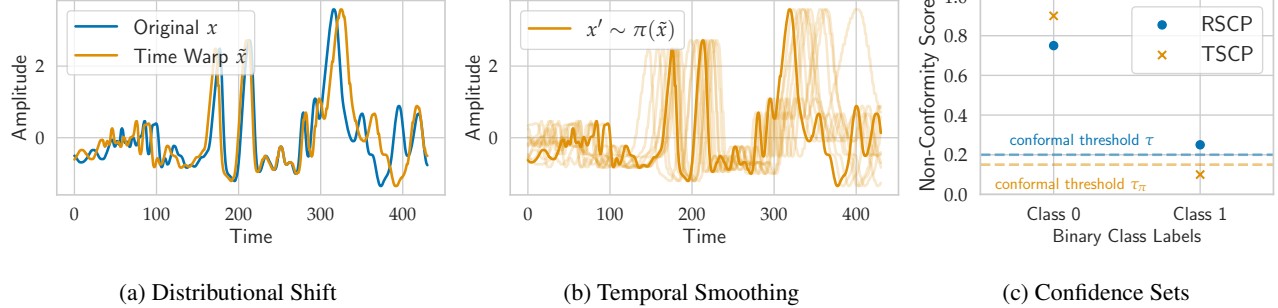

| (a) Distributional Shift | (b) Temporal Smoothing | (c) Confidence Sets |

Figure 1: Visual representations showcasing different stages of a Temporal Smooth Conformal Predictor (TSCP) with a sample from the UCR-TS dataset [Dau et al., 2018]. In (a) we display the distributional shift where the original signal $x$ can be manipulated via time warping $\tilde{x}$. In (b) we represent the smoothing versions of $x'$ derived from a temporal shifts around $\tilde{x}$. While in (c) we illustrate non-conformity scores, highlighting that TSCP obtains more precise confidence sets with respect than the Randomly Smoothed Conformal Predictor (RSCP) Gendler et al. [2021].

series are subjected to different, practically relevant transformations such as time warping, window warping, scaling or jitter. Since different domains can be modeled, i.e., experiment parameters which introduce domain shifts are known, several data sets are recorded. First, we show how CP naturally generalizes RS and use the resulting algorithm to show robustness on the use case tailored perturbations. Second, we utilize the available domains during training to bound the performance of the algorithm on unseen test domains under loose assumptions Park et al. [2019, 2020, 2022a,b]. In Figure 1, we show a visual representation of our framework. By applying temporal transformations to smooth the sample, we enhance the certainty of the predicted confidence set. As observed, the output demonstrates increased confidence for class 0, leading to a reduction in the size of the predicted confidence set.

In summary, our core contributions are:

- Generalizes RS to handle various transformations in time series classification, like time, window and magnitude warping.

- Introduce a *Temporal Smooth Conformal Predictor* (TSCP) that employees temporal-shift transformations to smooth the input and demonstrates CP guarantees robustness against adversarial attacks and perturbations.

- Experimental evaluation of TSCP on a open-source time series dataset and on a real-world application, demonstrating the effectiveness in terms of robustness certification and empirical risk minimization.

## 2 RELATED WORK

**Conformal Prediction** In CP [Vovk et al., 1999, 2005, Papadopoulos, 2008], the main objective is to generate a set of predictions that will likely include the correct label, an approach known as *marginal coverage*. This is achievable

under the assumption that the training and test examples are exchangeably distributed, as is the case for identical and independent distributed (iid) data. Nevertheless, this assumption does not hold in general in cases of *distribution shift*, where the test data's distribution differs from the training data. In such scenarios, previous research [Tibshirani et al., 2019, Cauchois et al., 2020, Gibbs and Candes, 2021, Park et al., 2022a] offers a solution, maintaining marginal coverage by adjusting for the differences in likelihood between training and test datasets. Despite these advancements, there remains a lack of reliable assurances regarding the robustness of the predictions.

**Randomized Smoothing** RS is a technique used to enhance the robustness of machine learning models by introducing random Gaussian noise to the input data and aggregating the predicted outputs [Cohen et al., 2019, Salman et al., 2019]. The prediction is robust against all disturbances within a norm-ball of radius $R$. Subsequent studies have expanded the robustness guarantees of RS to include semantic and real-world transformations [Fischer et al., 2020, Li et al., 2021, Hao et al., 2022]. Additionally, Yoon et al. [2022] extended its application to time series data, demonstrating its effectiveness in temporal shifting the input data.

Recently, Gendler et al. [2021] combined RS with CP, resulting in a better and robust coverage against adversarial input manipulations, denoted as RSCP. However, the strategy for addressing extreme-case Gaussian perturbations can diminish the baseline performance (assessment using only unperturbed inputs) of the CP method. For example, the size of the prediction set might be excessively large even for unperturbed and straightforward-to-classify inputs. To overcome these challenges, Ghosh et al. [2023] proposed an adjustment on the construction of the prediction set based on a two thresholds framework. Even if the results surpassed RSCP in performance (coverage and set-size) the guarantees are only probabilistic in practice and the approach is limited

to transformations that follow a normal distribution.

# 3 PRELIMINARIES

Let us consider $x \in \mathcal{X}$ as a multi-variate time series signal where $\mathcal{X} \subseteq \mathbb{R}^{c \times T}$, with $c$ denoting the number of channels and $T$ specifying the signal's window size. Here, we are examining a *soft* classifier, denoted as $F \colon \mathbb{R}^{c \times T} \to \mathbb{P}(\mathcal{Y})$, where $\mathbb{P}(\mathcal{Y})$ is the set of probability distributions over $\mathcal{Y}$, and $\mathcal{Y} = \{0, 1, \ldots, K\}$ is the collection of classification labels. Thus, a soft classifier assigns each data point a distribution over classes, rather than just assigning it to a class. It is possible to convert any soft classifier $F$ into a hard classifier $f$ by mapping $f(x) \overset{\text{def}}{=} \arg\max_{y \in \mathcal{Y}} F(x)_y$. In addition, we denote with $I$ the identity matrix, with $\mathcal{N}(0, \sigma^2 I)$ the standard normal distribution and with $\mathcal{U}(a, b)$ the uniform integer distribution where $a, b \in \mathbb{Z}, a < b$.

## 3.1 CONFORMAL PREDICTION (CP)

Introduced by Vovk et al. [1999, 2005] and Papadopoulos [2008], CP offers an intuitive approach to produce prediction sets that achieve a user defined confidence. In essence, given $n$ training samples $(x^{(i)}, y^{(i)})_{i=1}^{n}$, the objective is to predict the label $y^{(n+1)}$ for a test point $x^{(n+1)}$. Assuming the training and test samples come from an exchangeable source (i.i.d. distribution), CP methods create a prediction set $\mathcal{C}(x^{(n+1)})$ that is likely to include the test label $y^{(n+1)}$ with a specified coverage, such as 90% or 95%. Formally, this is expressed as:

$$\mathbb{P}[y^{(n+1)} \in \mathcal{C}(x^{(n+1)})] \geq 1 - \alpha, \qquad (1)$$

where $\alpha$ represents the chosen error rate. Notably, this probability considers all training samples and the test point $x^{(n+1)}$ and is also known as marginal coverage. CP's core principle involves training a classifier on the dataset and subsequently assigning *non-conformity scores* to validation data. Generally, lower prediction errors correspond to more concise and informative prediction sets.

**(Non) Conformity Score** The process initiates by dividing the training data into two distinct subsets: (i) a primary training set, denoted as $\mathcal{D}_{tr}$ within the range of $1, \ldots, n$, and (ii) a calibration set $\mathcal{D}_{cal}$, which is the remainder of the range after subtracting $\mathcal{I}_{tr}$. A soft classifier, represented as $F(x)$ spanning $[0, 1]^K$, is then trained on the primary set to determine the conditional probabilities of each class $\mathbb{P}[y|x]$ for every $y \in Y$. When using deep network classifiers, the subject of this study, this is typically the outcome of the softmax layer. Subsequently, a score function $S \colon \mathcal{X} \times \mathcal{Y} \to \mathbb{R}_{\geq 0}$ produces a *(non) conformity score*, $S^{(i)} = S(x^{(i)}, y^{(i)})$ for each point in the calibration set. This score evaluates the coherence between the model's prediction $F(x)$ and the actual label $y$, where a smaller score denotes a closer match.

**Definition 3.1** (Conformal Prediciton Set). *Given the desired coverage level $1 - \alpha$, a prediction set $\mathcal{C}$ for a new test point $x^{(n+1)}$ is defined as:*

$$\mathcal{C} = \left\{ y \in \mathcal{Y} : S(x^{(n+1)}, y) \leq Q_{1-\alpha}(\{S^{(i)}\}_{i \in \mathcal{D}_{cal}}) \right\}, \tag{2}$$

*where $Q_{1-\alpha}(\{S^{(i)}\}_{i \in \mathcal{D}_{cal}})$ is defined as the $(1 - \alpha)(1 + 1/(1+|\mathcal{D}_{cal}|))$-th empirical quantile of $\{S^{(i)}\}_{i \in \mathcal{D}_{cal}}$.*

In other words, Eq. 2 involves scanning through all potential labels $y \in \mathcal{Y}$ and adding to $\mathcal{C}(x^{(n+1)})$ those predicted labels $y$ whose scores $S(x^{(n+1)}, y)$ are lower than the majority of calibration scores $S(x^{(i)}, y^{(i)}), \forall i \in \mathcal{D}_{cal}$ [Vovk et al., 2005].

## 3.2 PAC PREDICTION SET

Our goal is to find conformal sets that are not only as compact as possible, but also highly reliable, adhering to the principle of being *probably approximately correct* (PAC) [Valiant, 1984]. Formally, considering an *algorithm* $\mathcal{A}$ that takes a set of calibration data $\mathcal{D}_{cal} \subset \mathcal{D}_{tr}$ and generates a CP set $\mathcal{C}$. Given $\gamma, \xi \in (0, 1)$, we consider $\mathcal{A}$ is PAC if:

$$\mathbb{P}_{\mathcal{D}_{cal} \sim \mathcal{D}_{tr}} [L_{\mathcal{D}_{cal}}(\mathcal{C}) \leq \xi \mid \mathcal{C} = \mathcal{A}(\mathcal{D}_{cal})] \geq 1 - \gamma, \quad (3)$$

where $L_{\mathcal{D}_{cal}}(\mathcal{C}) = \mathbb{P}_{(x,y) \sim \mathcal{D}_{cal}} [y \notin \mathcal{C}(x)]$ is the true error. The challenge lies in developing an algorithm $\mathcal{A}$ that not only meets the PAC criteria but also constructs confidence sets $\mathcal{C}(x)$ that are, on average, as minimal as possible. In the context of machine learning, Park et al. [2019, 2020] proposes to construct $\mathcal{C}$ by parametrizing it with a scalar $\tau \in \mathcal{T} \subseteq \mathbb{R}_{\geq 0}$ as:

$$\mathcal{C}_\tau = \{y \in \mathcal{Y} : S(x, y) \geq \tau\}, \qquad (4)$$

where $\tau$ represents the threshold which controls the trade-off between size and expected error. Intuitively, they formulate this challenge into an *empirical risk minimization* framework, where the objective is to minimize the size of the predicted confidence sets. In practice, the goal is to find the maximum threshold value $\hat{\tau}$ such that the empirical error $\hat{L}_{\mathcal{D}_{cal}}(\mathcal{C}) = \sum_{(x,y) \in \mathcal{D}_{cal}} \mathbb{1}(y \notin \mathcal{C}_\tau(x))$ remains within a certain confidence interval. Formally, this is expressed as:

$$\hat{\tau} = \max_{\tau \in \mathcal{T}} \left\{ \tau : \hat{L}_{\mathcal{D}_{cal}}(\mathcal{C}_\tau) \leq k(m, \xi, \gamma) \right\}, \qquad (5)$$

where the right-hand side of the inequality is the confidence level $k$ derived from the Binomial distribution as follows:

$$k(m, \xi, \gamma) = \max_{k \in \mathbb{N}_0} \left\{ k : \sum_{i=0}^{k} \binom{m}{k} \xi^i (1 - \xi)^{m-i} < \gamma \right\}. \tag{6}$$

This approach is conceptually linked to the idea that the average loss behaves like a Binomial distribution. By setting $\hat{\tau}$ in this manner, we aim to minimize the size of the

confidence sets while ensuring that the empirical error stays within acceptable probabilistic bounds, thereby adhering to the PAC guarantee of Eq. 3.

## 3.3 SMOOTHED CONFORMAL PREDICTION

Initially introduced by Cohen et al. [2019] and Salman et al. [2019], randomized smoothing computes the $\ell_2$-norm certificates around an input sample $x$ by counting which class is most likely to be returned when $x$ is perturbed by isotropic Gaussian noise.

**Definition 3.2** (Smooth Classifier). *Given a soft classifier $F$, randomized smoothing considers a smooth version of $F$ defined as:*

$$G(x) \stackrel{def}{=} \underset{\delta \sim \mathcal{N}(0,\sigma^2 I)}{\mathbb{E}} \left[F(x + \delta)\right], \qquad (7)$$

*where $\sigma > 0$ represents the standard deviation.*

Cohen et al. [2019] demonstrated that $G$ is robust to perturbations of radius $R$, where the radius $R$ is defined as the difference in probabilities between the most likely class and the second most likely class. Contrary to other formal verification methods, randomized smoothing does not make any assumptions regarding the model's properties, allowing certification to be scaled to larger and more complex networks.

**Smoothed Score** Interestingly, this inherent robustness offered by randomized smoothing served as an additional layer to address challenges in conformal predictions [Gendler et al., 2021, Ghosh et al., 2023]. Sets formed by the basic conformal method may not ensure accurate coverage, especially when real-world data breaches the exchangeability assumption due to frequent distribution shifts [Tibshirani et al., 2019, Cauchois et al., 2020, Gibbs and Candes, 2021]. In a recent work, Gendler et al. [2021] introduced a *smooth* version of the original non-conformity score obtained by averaging the value of $S(x + \delta, y)$ over many independent samples.

**Definition 3.3** (Smooth Score). *Let $S : \mathcal{X} \times \mathcal{Y} \to \mathbb{R}_{\geq 0}$ be a scoring function. We define the smoothed score function as:*

$$\tilde{S}(x,y) \stackrel{def}{=} \Phi^{-1} \left( \underset{\delta \sim \mathcal{N}(0,\sigma^2 I)}{\mathbb{E}} \left[S(x + \delta, y)\right] \right), \quad (8)$$

*where $\Phi^{-1}$ is the inverse of the cumulative distribution function (quantile) of the standard normal distribution.*

As shown in Salman et al. [2019] and Gendler et al. [2021], the local Lipschitz continuity derived from randomly smoothing the prediction, sets an upper-bound for the conformal score:

$$\tilde{S}(\tilde{x}^{(n+1)}, y) \leq \tilde{S}(x^{(n+1)}, y) + \frac{R_\delta}{\sigma}, \qquad (9)$$

where it holds for every $y \in \mathcal{Y}$. If we consider a distance metric between $\tilde{x}$ and $x$, such that $d(\tilde{x}, x) \leq \epsilon$, with $\epsilon > 0$, than for a Gaussian distribution $\delta \sim \mathcal{N}(0, \sigma^2 I)$ the radius $R_\delta$ corresponds to $\|\epsilon\|_2$.

# 4 GENERALIZED ROBUSTNESS VIA CONFORMALIZED RANDOMIZED SMOOTHING

This section introduces the methods and ideas related to smooth conformal predictions for robustness certification. For ease of explanation, in this section, we will regard the input $x \in \mathbb{R}^T$ as one-variate signal.

## 4.1 GENERALIZED SMOOTHED CLASSIFIER

Following previous works on image classifiers [Li et al., 2021, Hao et al., 2022], we introduce a smooth classifier by randomly transforming inputs with parameters sampled from a smoothing distribution. An important aspect is that even if the definition is general and applies towards any transformation, our focus is on time series augmentations.

Let us consider a transformation $\phi : \mathcal{X} \times \mathcal{Z} \to \mathcal{X}$ which produces a unique augmented version of the time series $x$, leading to a distinct $\tilde{x}$. In this notation, $\mathcal{Z}$ represents the set of parameters. In App. A, we define the set of time series transformations $\phi$ considered in this work.

**Definition 4.1** (Generalized Smoothed Classifier). *Let $\phi : \mathcal{X} \times \mathcal{Z} \to \mathcal{X}$ be a transformation, $\pi \sim \mathcal{D}_\pi$ a random variable taking values in $\mathcal{Z}$ and let $F : \mathbb{R}^T \to \mathbb{R}$ be a soft classifier. We define the $\phi$-smoothed version $G_\phi : \mathcal{X} \to \mathbb{P}(\mathcal{Y})$ of $F$ as:*

$$G_\phi(x) \stackrel{def}{=} \underset{\pi \sim \mathcal{D}_\pi}{\mathbb{E}} \left[F(\phi(x, \pi))\right]. \qquad (10)$$

Drawing from Theorem 1 in Li et al. [2021], it is possible to establish a robustness certificate for the classifier $G_\pi$ that employs a $\phi$-smoothing technique. In Sec. 5.2, we discuss the robustness guarantees for a specific set of time series transformations. In general, take an input $x \in \mathcal{X}$ and a random variable $\pi \in \mathcal{Z}$. The soft classifier $F$ assesses that $\tilde{x} = \phi(x, \pi)$ is likely to be in class $y_A$ with a probability of at least $p_A$, and the likelihood of it being in the second most probable class does not exceed $p_B$. To establish a robustness certificate, one must identify a set of perturbation parameters $\mathcal{Z}_\lambda \subseteq \mathcal{Z}$ and to ensure that for all perturbations $\lambda \in \mathcal{Z}_\lambda$, the classifier $G_\phi$'s output for $\phi(x, \lambda)$ remains identical to its output for $x$, i.e. $G_\phi(\phi(x, \lambda)) = G_\phi(x)$.

Lastly, we establish a $\phi$-smoothed conformal score for $G_\phi$. Unlike the approach in Gendler et al. [2021], we incorporate a broader range of transformations.

**Definition 4.2** (Generalized Smoothed Score). *Let $\phi : \mathcal{X} \times \mathcal{Z} \to \mathcal{X}$ be a transformation, $\pi \sim \mathcal{D}_\pi$ a random variable taking values in $\mathcal{Z}$ and $S : \mathcal{X} \times \mathcal{Y} \to \mathbb{R}_{\geq 0}$ a scoring function. We define the $\phi$-smoothed score function as:*

$$S_\phi(x,y) \stackrel{def}{=} Q\left(\underset{\pi \sim \mathcal{D}_\pi}{\mathbb{E}}\left[S(\phi(x,\pi),y)\right]\right), \quad (11)$$

*where $Q : [0,1] \to \mathbb{R}$ represents the quantile function.*

## 4.2 ROBUSTNESS GUARANTEES FOR CONFORMAL PREDICTIONS UNDER GENERAL TRANSFORMATIONS

In the context of domain generalization, where the assumption of i.i.d. data no longer applies, it becomes crucial to estimate the potential shift between a baseline smooth score and one that comes from a different domain or has been attacked. This estimation is necessary to effectively bound the *distribution shift*. This approach extend the setting of Gendler et al. [2021], to a broader range of input transformations. We approach this by considering a non-conformity score function $S_\phi$ as defined in Def. 4.2, which allows us to gauge the extent of change brought on by a transformation function to $x^{(n+1)}$. Our task is to ensure that $S_\phi$ complies with the condition:

$$S_\phi(\tilde{x}^{(n+1)},y) \leq S_\phi(x^{(n+1)},y) + R_\pi, \ \forall y \in \mathcal{Y}, \quad (12)$$

where $\tilde{x}^{(n+1)} = \phi(x^{(n+1)},\pi)$ and $R_\pi$ is a constant connected to $\pi$, fulfilling the criteria that $R_{\pi_1} \leq R_{\pi_2}$ if $\pi_1 \leq \pi_2$, and $R_\pi$ is zero when $\pi$ is zero.

The exact derivation of the robustness radius $R_\pi$ depends on the transformation considered. Strictly speaking, our objective is to verify the robustness in response to a transformation $\phi$ that can be effectively addressed by $\psi$, and this verification pertains to transformation parameters contained within the set $\mathcal{Z}_\lambda \subseteq \mathcal{Z}$. To achieve this, we begin by selecting a set of parameters $\{\lambda_j\}_{j=1}^N$ from the parameter space $\mathcal{Z}_\lambda$. We then apply these parameters to transform the input data, generating a collection of transformed inputs $\{\phi(x,\lambda_j)\}_{j=1}^N$. Next, we utilize the classifier (which has been enhanced with the transformable transformation $\psi$) to calculate the class probabilities for each of these transformed inputs. Following Li et al. [2021, Corollary 2], if the guaranteed robustness radius $R_\pi$, defined as:

$$R_\pi \stackrel{def}{=} \frac{\sigma}{2} \min_{1 \leq j \leq N}\left(\Phi^{-1}(p_A^{(j)}) - \Phi^{-1}(p_B^{(j)})\right) \quad (13)$$

for differentially resolvable transformations is greater than the maximum interpolation error:

$$M_{\mathcal{Z}_\lambda} = \max_{\lambda \in \mathcal{Z}_\lambda} \min_{1 \leq j \leq N} \|\phi(x,\lambda) - \phi(x,\lambda_j)\|_2 < R_\pi \quad (14)$$

then the it is guaranteed that $\forall \lambda \in \mathcal{Z}_\lambda$, the smooth classifier will continue classify the original predicted class. Practically,

given a transformation $\phi$, if the conditions identified in Table 1 are satisfied, $S_\phi$ provides a tight certified distance $R_\pi$ that satisfies Eq. 12.

In this context, $R_\pi$ is instrumental in linking the observed score $S_\phi(\tilde{x}^{(n+1)},y)$ with the unobserved score $S_\phi(x^{(n+1)},y)$ for any given $y \in \mathcal{Y}$. Leveraging this relationship, we construct a prediction set $\mathcal{C}_\pi(\tilde{x}^{(n+1)})$ resilient to input transformations with bounded deviation, following a decision rule:

$$\left\{y \in \mathcal{Y} : S_\phi(\tilde{x}^{(n+1)},y) \leq Q_{1-\alpha}(\{S_\phi^{(i)}\}_{i \in \mathcal{D}_{cal}}) + R_\pi\right\}, \quad (15)$$

where $S_\phi^{(i)}$ is defined as $S_\phi(x^{(i)},y^{(i)})$. This approach diverges from the standard split conformal method of Eq. 2, as our prediction set is derived by comparing the test score against an elevated threshold $Q_{1-\alpha} + R_\pi$. This adjustment is dependent on both the magnitude of the transformation and the robustness of $S_\phi$, implying that a larger disturbance necessitates a higher threshold increase, while a more resilient $S_\phi$ requires a smaller increase.

**Theorem 4.1.** *Assume a set of samples $\{(x^{(i)}, y^{(i)})\}_{i=1}^{n+1}$ that are exchangeably drawn from an unknown distribution $\mathcal{D}_{xy}$. Let $\phi : \mathcal{X} \times \mathcal{Z} \to \mathcal{X}$ be a differentially resolvable transformation, let $\mathcal{Z}_\lambda \subseteq \mathcal{Z}$, $\{\lambda_j\}_{j=1}^N$ be a set of perturbation parameters and let $G : \mathcal{X} \to \mathbb{P}(\mathcal{Y})$ be a smooth classifier as in Def. 3.2 that predicts $y_A \in \mathcal{Y}$ given $x$ (i.e. $G(y_A \,|\, x)$ where $x = x^{(n+1)}$). If for any $j$, $G(x)$ has class probabilities that satisfy:*

$$G(y_A \,|\, \phi(x,\lambda_j)) \geq p_A^{(j)} \geq p_B^{(j)} \geq \max_{y \neq y_A} G(y \,|\, \phi(x,\lambda_j)), \quad (16)$$

*and Eq. 14 holds, then, the prediction set $\mathcal{C}_\pi$ as defined in Eq. 15 will satisfy the following probability:*

$$\mathbb{P}[y^{(n+1)} \in \mathcal{C}_\pi(\phi(x^{(n+1)},\pi))] \geq 1 - \alpha. \quad (17)$$

Proof is given in App. B. Thus, we assert that the prediction set $\mathcal{C}_\pi(\tilde{x}^{(n+1)})$ will include the unknown target label $y^{(n+1)}$ with a probability of at least $1 - \alpha$, regardless of the distribution $\mathcal{D}_{xy}$, sample size $n$, the score function $S_\pi$ adhering to Eq. 12, and the magnitude of adversarial perturbation $\pi$ generated by any attack algorithm.

## 4.3 BOUNDING THE DOMAIN GENERALIZATION

In this section, we broaden our examination to include the PAC theory and sketch guarantees for the *Generalized Smoothed Classifier* to comply with the PAC criteria outlined in Eq. 3.

Following Park et al. [2020, 2022a], the goal is to find an upper bound $\bar{\xi}(k; m, \gamma) \in [0,1]$ on the true success probability $\mu$, constructed from a sample $k \sim \text{Binom}(m, \mu)$, which

holds with probability at least $1 - \gamma$, where the probability mass function is defined as:

$$\mathbb{P}_B(k \mid m, \xi) = \sum_{i=0}^{k} \binom{m}{k} \xi^i (1 - \xi)^{m-i}. \quad (18)$$

The PAC guarantees is expressed as:

$$\mathbb{P}_{k \sim \text{Binom}(m,\mu)}[\mu \leq \bar{\xi}(k \mid m, \gamma)] \geq 1 - \gamma, \quad (19)$$

where the upper bound $\bar{\xi}$ is defined as:

$$\bar{\xi}(k \mid m, \gamma) \stackrel{\text{def}}{=} \inf_{\xi \in [0,1]} \{\xi \; : \; \mathbb{P}_B(k \mid m, \xi) \leq \gamma\} \cup \{1\}. \quad (20)$$

In other words, the true error $L_{\mathcal{D}_{cal}}(\mathcal{C})$ is bounded by the upper bound $\bar{\xi}(\hat{L}_{\mathcal{D}_{cal}}(\mathcal{C}) \mid m, \gamma)$ with probability at least $1 - \gamma$.

In our analysis, we consider the conformal set $\mathcal{C}_\pi$ of Eq. 15 and bound the generalization error by adjusting the estimated threshold $\hat{\tau}$, defined in Eq. 5, by the robustness $R_\pi$ radius of the smooth classifier. To do so, we consider a mapping function $\psi_\pi : \mathcal{X} \times \mathcal{Y} \to \mathbb{R}$ that incorporates the score function $S_\phi$ and the robustness radius $R_\pi$, and encodes the prediction set condition into a binary classification framework:

$$\psi_\pi(\tilde{x}, y) \stackrel{\text{def}}{=} S_\phi(\tilde{x}, y) - Q_{1-\alpha}(\{S_\phi^{(i)}\}_{i \in \mathcal{D}_{cal}}) - R_\pi, \quad (21)$$

where $\tilde{x} = \tilde{x}^{(n+1)} = \phi(x^{(n+1)}, y)$. Thus, let us define a binary function $M_\tau(t) = \mathbb{I}[t \leq 0]$, such that we can re-write the confidence set $\mathcal{C}_\tau$ as:

$$\mathcal{C}_{\pi,\tau}(\tilde{x}^{(n+1)}) = \left\{ y \in \mathcal{Y} : M_\tau(\psi(\tilde{x}^{(n+1)}, y)) = 1 \right\}. \quad (22)$$

Thus, the PAC bound for the binary classifier $M_\tau$ will then imply a PAC bound for the confidence set predictor $\mathcal{C}_{\pi,\tau}$, ensuring that the prediction set adheres to the desired probability bounds. This means we need to establish a PAC bound for $M_\tau$ under the modified encoding that incorporates $S_\phi$, $Q_{1-\alpha}$, and $R_\pi$. In practice, we can obtain an empirical threshold $\hat{\tau}_\pi$ defined as:

$$\hat{\tau}_\pi = \sup_{\tau \in \mathcal{T}} \left\{ \tau \; : \; \hat{L}_{\mathcal{D}_{cal}}(\mathcal{C}_{\pi,\tau}) \leq k(m, \xi, \gamma) \right\}, \quad (23)$$

which depends on the distribution $\pi$ and on the confidence level $k$ defined in Eq. 6.

# 5 CERTIFIED ROBUSTNESS FOR TEMPORAL TRANSFORMATIONS

In this section, we introduce a temporal transformation for stretching and compressing time series. Building upon previous work [Li et al., 2021], we establish proven robustness guarantees. We conclude by presenting our *Temporal Smooth Conformal Predictor* (TSCP).

## 5.1 RANDOM TIME WARPING

In practice, the time warping transformation $\phi$ is centered around a randomly chosen warp point $p \in \mathbb{N}$ with $0 < p < T$, and involves stretching and compressing different sections of $x$ while preserving its overall length. The time warping process is characterized by two key parameters: $w_l, w_r \in \mathbb{N}$, representing the warp factors for the left and right sides of $p$, respectively. The warp factor $w_l$ is selected randomly from a uniform distribution in the range $(0, \lceil \theta \cdot T \rceil)$, where $0 < \theta < 1, \theta \in \mathbb{R}$ denotes the warp size. The warp factor $w_r$ is then calculated to ensure a balanced warp, maintaining the length of $T$. Formally, this relationship is expressed as:

$$w_r = w_l \cdot \frac{p}{T - p}. \quad (24)$$

For each index $i$ in the original time series $x$, the corresponding index $\tilde{i}$ in the warped time series $\tilde{x}$ is determined based on $w_l$, $w_r$, and $p$. Specifically, the warped indices are computed as follows:

$$\tilde{i} = \begin{cases} i + \lceil w_l \cdot \frac{p-i}{p} \rceil & \text{for } i < p, \\ i - \lceil w_r \cdot \frac{i-p}{T-p} \rceil & \text{for } i \geq p. \end{cases} \quad (25)$$

Finally, the warped time series $\tilde{x}$ is constructed by mapping each value $t_i$ from the original time series $x$ to the corresponding warped index $\tilde{i}$. In Alg. 1, we synthesize the overall procedure.

---

**Algorithm 1** Random Time Warping of a Time Series

---

1: **procedure** RANDTIMEWARP$(x, \theta)$
2:     **initialize:**$p \sim \mathcal{U}[1, T - 1]$, $w_l \sim \mathcal{U}[1, \lceil \theta \cdot T - 1 \rceil]$
3:     $\tilde{x} \leftarrow x$; $w_r \leftarrow w_l \cdot \frac{p}{T-p}$
4:     **for** $i = 0$ **to** $T - 1$ **do**
5:         **if** $i < p$ **then**
6:             $\tilde{i} \leftarrow i + \lceil w_l \cdot \frac{p-i}{p} \rceil$
7:         **else**
8:             $\tilde{i} \leftarrow i - \lceil w_r \cdot \frac{i-p}{T-p} \rceil$
9:         **end if**
10:         $\tilde{x}_{\tilde{i}} \leftarrow x_i$
11:     **end for**
12:     **return** $\tilde{x}$
13: **end procedure**

---

**Numerical Complexity** Here, we discuss the computational complexity of our time warping augmentation method in relation to earlier studies [Le Guennec et al., 2016, Um et al., 2017, Iwana and Uchida, 2021b]. Traditionally, time warping involves creating a cubic spline using a series of knots, a process that typically requires solving a tridiagonal system of equations. Once constructed, this spline is applied across the time series. The complexity of this method is primarily dictated by the number of knots, $I$, and the time series length, $T$, resulting in an overall linear complexity of $\mathcal{O}(I + T)$. In contrast, our proposed method adopts a

Table 1: Certified robustness radii for resolvable and differentially resolvable time series transformations.

| Type | Transformation ($\pi$) | Distribution | Certified Robustness Radius ($R_\pi$) |
|------|------------------------|--------------|----------------------------------------|
| Resolvable | Jitter | $\delta \sim \mathcal{N}(0, \sigma^2 I)$ | $\frac{\sigma}{2}\left(\Phi^{-1}(p_A) - \Phi^{-1}(p_B)\right)$ |
| | Scaling | $\delta \sim \mathcal{N}(1, \sigma^2 I)$ | $\frac{1}{2}\left(\Phi^{-1}(p_A) - \Phi^{-1}(p_B)\right)$ |
| Diff. Resolvable | Magnitude-warp | $u \sim \mathcal{N}(1, \sigma^2 I)$ | $\frac{\sigma}{2}\min_{1 \le j \le N}\left(\Phi^{-1}(p_A{}^{(j)}) - \Phi^{-1}(p_B{}^{(j)})\right)$ |
| | Time warp | $p \sim \mathcal{U}[0, T]$ | $\frac{\sigma}{2}\min_{1 \le j \le N}\left(\Phi^{-1}(p_A{}^{(j)}) - \Phi^{-1}(p_B{}^{(j)})\right)$ |
| | Window-warp | $p \sim \mathcal{U}[0, T]$ | $\frac{\sigma}{2}\min_{1 \le j \le N}\left(\Phi^{-1}(p_A{}^{(j)}) - \Phi^{-1}(p_B{}^{(j)})\right)$ |

more straightforward approach. It primarily consists of a loop that runs through the time series, executing one simple arithmetic operation for each element. This results in a linear complexity of $\mathcal{O}(T)$, making it $I$-times more efficient, especially when the number of knots $I$ in the cubic spline method is significantly large.

## 5.2 ROBUSTNESS RADII

Similarly to Li et al. [2021], we categorize the transformations into two types: *resolvable* and *differentiably resolvable*. In Table 1, we report the certified radius for each individual transformation considered.

As previously discussed in Sec. 4.2, we consider the methods for computing a tight and scalable upper bound $M$ for the interpolation error $M_{\mathcal{Z}_\lambda}$ in resolvable and differentially resolvable time series transformations. The process begins by selecting a subset of transformation parameters $\{\lambda\}_{j=1}^N$ from $\mathcal{Z}_\lambda$, and applying these parameters to transform the input, resulting in a set of transformed inputs $\{\phi(x, \lambda_j)\}_{j=1}^N$. Subsequently, the class probabilities for each of these transformed inputs are calculated using a classifier that has been smoothed with the transformation $\psi$. The underlying principle is that if each parameter $\lambda_j$ in $\mathcal{Z}_\lambda$ is sufficiently close to one of the sampled parameters, then the classifier can be considered robust against any parameters from the set $\mathcal{Z}_\lambda$. This forms a crucial part of the methodology for ensuring both the accuracy and scalability of the upper bound $M$ in relation to the certification of transformations, particularly those involving interpolation errors.

**Jitter** This method aligns with the application and bounds derivation associated with smooth classifiers as formerly described in Cohen et al. [2019], Salman et al. [2019]. The convolution of a Gaussian process with the input signal, formerly recognized as the Weierstrass transform [Bilodeau, 1962], provides an alternative yet equivalent perspective on the certified robustness assurances for predictions [Salman et al., 2019].

**Scaling** As one might expect, scaling a time series is quite like adjusting the contrast in an image. To determine a guaranteed robustness radius from this, we can calculate the probability of the leading predicted class, denoted as $p_A$,

and the next closest class, $p_B$, using Monte-Carlo sampling (refer to Corollary 7; Appendix D in Li et al. [2021]). The robustness radius is then determined by taking half the difference between the quantiles of these two probabilities.

**Magnitude & window warping** In the context of magnitude and window warping computing an upper bound on the interpolation error is related to find the maximum value of the derivative of the cubic spline interpolation. In general, we can calculate an upper bound for interpolation error in transformations, using stratified sampling Li et al. [2021]. An interval of transformation parameters, $\mathcal{Z}_\lambda = [a, b]$, is divided uniformly into $N$ parameters, $\lambda_i$. For these parameters, functions $g_i : [a, b] \to \mathbb{R}_{\ge 0}$, representing squared $\ell_2$ interpolation error between transformed samples, are defined as:

$$\lambda \to g_i(\lambda) \stackrel{\text{def}}{=} \|\phi(x, \lambda) - \phi(x, \lambda_i)\|_2^2. \quad (26)$$

The goal is to find an upper bound, $M_i$, for each sub-interval $[\lambda_i, \lambda_{i+1}]$ such that:

$$M_i \ge \max_{\lambda_i \le \lambda \le \lambda_{i+1}} \min\{g_i(\lambda), g_{i+1}(\lambda)\}. \quad (27)$$

This leads to an overall upper bound $\sqrt{M} \stackrel{\text{def}}{=} \max_{1 \le i \le N-1} \sqrt{M_i}$, which is valid for the entire interval $\mathcal{Z}_\lambda$.

Second-level sampling (n) is conducted within each sub-interval $[\lambda_i, \lambda_{i+1}]$, dividing them uniformly into n parameters, $\{\gamma_{i,j}\}_{j=1}^n$. If we have that $L$ is a global Lipschitz constant for all functions $\{g_i\}_{i=1}^N$, a closed-form expression for $M_i$ can be derived. In App. C, we compute the global derivative and bound it by a Lipschitz constant for a cubic spline interpolation. With a global Lipschitz constant $L$ for all $g_j$ functions, a closed-form expression for $M_j$ can be derived [Li et al., 2021]. This methodology shows that increasing the number of first-level ($N$) or second-level ($n$) samples results in a tighter upper bound on interpolation error.

**Time warping** In the context of time warping, $\phi$ alters the indices of the time series based on the parameters $w_l$ and $w_r$, with the transformation centered around the point $p$. The derivative of the warping function $\phi$ essentially represents

the rate of change of the warped indices with respect to the original indices. Formally,

$$d\phi(i) = \begin{cases} 1 - \frac{w_l}{p} & \text{for } i < p, \\ 1 + \frac{w_r}{T-p} & \text{for } i \geq p. \end{cases} \quad (28)$$

Since $w_l \leq p \leq T$, the derivatives are always positive. Given that $w_l$ is selected in the range $(0, \lceil \theta \cdot T \rceil)$ and $w_r = w_l \cdot \frac{p}{T-p}$, we can compute the upper bounds for both derivatives.

### 5.3 TEMPORAL SMOOTH CONFORMAL PREDICTOR

---
**Algorithm 2** TSCP: Temporal Smooth Conformal Predictor
---
**Require:** target error rate $\alpha \in (0, 1)$, transformation $\phi$, budget $\sigma$, smoothing samples $N$, data split into training $\mathcal{D}_{tr}$ and calibration $\mathcal{D}_{cal}$ sets.
1: Train a classifier $F$ on $\mathcal{D}_{tr}$.
2: Compute generalized smoothed scores $\{S_\phi^{(i)}\}_{i \in \mathcal{D}_{cal}}$.
3: Compute the empirical quantile $Q_{1-\alpha}(\{S_\phi^{(i)}\}_{i \in \mathcal{D}_{cal}})$.
4: Given $\tilde{x}^{(i+1)}$, construct $\mathcal{C}_\pi(\tilde{x}^{(n+1)})$ as in Eq. 15.
---

In Alg. 2 we present our method. It is primarily designed to generate reliable predictions within a defined error range $\alpha$. It operates by considering a transformation function $\phi$, a budget constraints $\sigma$, and a number of smoothing samples $N$. The algorithm calculates generalized smoothed scores for the calibration dataset and determines the empirical quantile from these scores, aligning with the target error rate. The final step involves constructing a conformal prediction set for any new input, ensuring that the predictions adhere to the set error rate and maintain the required level of reliability.

## 6 EXPERIMENTS

In our investigation of time series classification, we compare the generalized smoothed classifier, as defined in Sec. 4.1 with the *vanilla* classifier. We dub the classifier *Temporal Smooth Conformal Predictor* (TSCP) since our main goal is to understand how smoothing the input with time series native perturbation (see App. A) affects the coverage and accuracy of CP. In the first experiment, we examine adversarial attacks to measure the robustness of TSCP against intentionally crafted disturbances. Next, we explore domain generalization to assess how well the models adapt to different operational settings. These experiments are structured to provide a clearer understanding of the relative advantages of TSCP in managing the dynamic and noisy conditions typical of vehicle operation, and to identify a suitable architecture for time series classification.

### 6.1 SETTINGS

In our analysis, we consider two datasets: the UCR time series classification archive Dau et al. [2018] composed of 128 time series datasets and an in-house dataset composed of 7 input signals resulting from vehicle sensors. After an initial binning, a single data point contains 500 time steps, resulting in a data snippet of $x \in \mathbb{R}^{7 \times 500}$, which is then classified into two classes $y \in \{0, 1\}$. As classifiers, we employ two distinct neural network architectures: a convolutional neural network (CNN) for the UCR datasets and a time series transformer for the in-house dataset. Additional details are reported in App. D.

**Coverage** In the context of conformal prediction, coverage is a pivotal metric that measures the accuracy of the predictive model's confidence intervals. Essentially, coverage represents the proportion of times the true labels fall within the prediction intervals generated by the model. Formally, the coverage can be expressed as

$$Coverage = \frac{1}{|\mathcal{D}_{test}|} \sum_{(x,y) \in \mathcal{D}_{test}} \mathbb{1}(y \in \mathcal{C}(x)), \quad (29)$$

where $\mathcal{C}(x)$ denotes the conformal prediction set generated by the model, $\mathbb{1}$ is the indicator function, which equals 1 if the condition $y \in C(x)$ is true, and 0 otherwise.

**Hardware Resources** The experiments were conducted utilizing a server with four NVIDIA A100 GPUs and an AMD EPYC 7542 32-Core CPU.

### 6.2 ADVERSARIAL ROBUSTNESS ON UCR

In this section, we explore the effectiveness of different classifiers in a white-box setting, focusing on their susceptibility to evasion attacks. Using Projected Gradient Descent (PGD) Carlini and Wagner [2017], we assess the robustness of three prediction techniques: CP Vovk et al. [2005], RSCP Gendler et al. [2021], and our method (TSCP). Our analysis delves into each method's accuracy, coverage, and prediction set-size under varying levels of adversarial perturbations. In this context, we performed 20 uniformly distributed PGD [Carlini and Wagner, 2017] attacks within $\epsilon \in [0, 0.1]$, for 40 iterations and a step size of $\epsilon \times 10^{-1}$. In the context of RSCP and TSCP, we consider 2000 samples. The primary objective is to shed light on how standard and smooth classification approaches behave when faced with increasingly intense adversarial samples, thereby evaluating their overall robustness.

Figure 2 presents top-1 accuracy, coverage and set-size of CP, RSCP, and TSCP under escalating $\ell_\infty$-norm perturbations, denoted by $\epsilon$, for one dataset of the UCR datasets [Dau et al., 2018]. We observe that RSCP and TSCP mantains

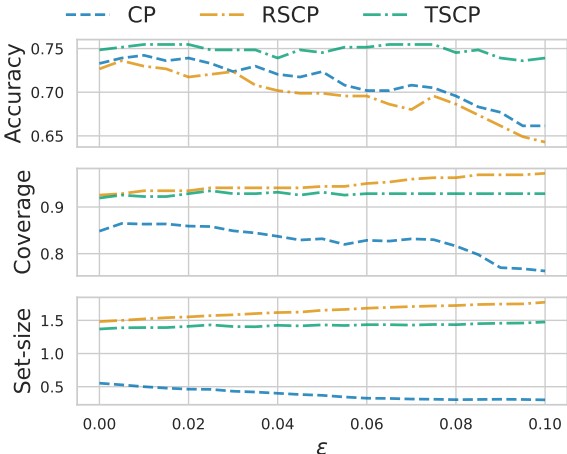

Figure 2: Top-1 accuracy, coverage and set-size comparison between CP [Vovk et al., 2005], RSCP [Gendler et al., 2021] and TSCP (our) under increasing $\ell_\infty$-norm adversarial perturbations with budget $\epsilon$ for the earth-quakes UCR dataset [Dau et al., 2018]. TSCP and RSCP have $\sigma = 0.2$.

higher coverage for increasing $\epsilon$ values, despite having different accuracy. This highlights the robustness of a smooth classifier against adversarial perturbations. However, the set-size increases slightly with increasing $\epsilon$, indicating larger prediction sets. Overall, TSCP tends to outperform in accuracy and coverage, but all methods show a degree of decline in performance with increasing adversarial budget.

In Table 2, we present a comprehensive comparison between CP, RSCP and TSCP across various datasets from the UCR archive (complete version in App. E). This assessment was limited to models that demostrate a minimum clean test top-1 accuracy of 70% or higher. We highlight the method that reaches a coverage as much close as possible to the target one 90% $(1-\alpha)$ and the target set-size which corresponds to 1. Notably, we prioritize methods that consistently achieve at least the desired coverage of 90%. This minimum threshold ensures a baseline level of robustness, as methods falling below this level are considered less reliable in the face of adversarial manipulation. The *Overall* row encapsulates the average performance and provides an aggregated view of the adversarial robustness across all datasets. In general, the performance of TSCP and RSCP are close each other.

We observe that despite augmenting the set-size, both the RSCP and TSCP methods successfully create conformal sets that adapt to changes in data distribution and maintain a coverage level above the target of 90%. For some of the data sets, both methods achieve 100% coverage. This might be attributed to the relatively small size of the validation and calibration sets (20% of the dataset, respectively). Few data points in the calibration and validation sets lead to overly cautious decision thresholds and fluctuations in coverage estimates. Therefore, while we observe 100% coverage in many cases, the actual coverage on unseen data might be

Table 2: Comparison analysis of CP [Vovk et al., 2005], RSCP [Gendler et al., 2021] and TSCP (our) across UCR [Dau et al., 2018] datasets. We consider an average of 20 uniformly distributed PGD [Carlini and Wagner, 2017] attack samples with $\epsilon \in [0, 0.1]$ and a target coverage of 90% $(\alpha = 0.1)$. RSCP and TSCP are augmented by $\sigma = 0.2$. The complete version is available in App. E.

| Dataset | Coverage | | | Set-Size | | |
|---|---|---|---|---|---|---|
| | CP | RSCP | TSCP | CP | RSCP | TSCP |
| ArrowHead | 78.7 | 97.5 | 99.4 | 0.83 | 2.41 | 2.53 |
| BME | 100.0 | 100.0 | 100.0 | 0.47 | 3.00 | 3.00 |
| Beef | 71.6 | 100.0 | 100.0 | 0.40 | 4.57 | 4.57 |
| BirdChicken | 78.2 | 100.0 | 100.0 | 0.74 | 2.00 | 2.00 |
| CBF | 97.2 | 100.0 | 100.0 | 2.14 | 3.00 | 3.00 |
| Car | 78.4 | 100.0 | 100.0 | 0.95 | 4.00 | 4.00 |
| Chinatown | 92.2 | 100.0 | 100.0 | 0.69 | 2.00 | 2.00 |
| CinC-ECG | 88.6 | 99.9 | 98.2 | 0.52 | 3.98 | 3.84 |
| Coffee | 56.4 | 99.1 | 53.9 | 0.61 | 1.94 | 0.99 |
| Cricket-X | 85.4 | 99.7 | 100.0 | 1.22 | 10.54 | 10.19 |
| Cricket-Z | 85.0 | 99.4 | 99.7 | 1.94 | 10.20 | 9.70 |
| Diatom Red. | 75.0 | 99.9 | 99.3 | 0.69 | 3.99 | 3.94 |
| Distal Age | 92.1 | 100.0 | 100.0 | 1.57 | 2.93 | 2.96 |
| Distal Correct | 93.3 | 99.8 | 100.0 | 1.72 | 1.97 | 1.97 |
| Distal TW | 90.6 | 100.0 | 100.0 | 2.17 | 4.48 | 4.35 |
| ⋮ | ⋮ | ⋮ | ⋮ | ⋮ | ⋮ | ⋮ |
| Toe Seg. 1 | 79.2 | 99.4 | 95.4 | 0.89 | 1.81 | 1.64 |
| Toe Seg. 2 | 93.9 | 98.7 | 95.6 | 0.66 | 1.45 | 1.38 |
| Trace | 90.1 | 100.0 | 100.0 | 0.68 | 3.14 | 3.18 |
| TwoLeadECG | 75.0 | 95.4 | 99.7 | 0.44 | 1.81 | 1.92 |
| Two-Patterns | 100.0 | 100.0 | 100.0 | 0.78 | 3.80 | 3.77 |
| UMD | 99.7 | 100.0 | 100.0 | 1.71 | 3.00 | 3.00 |
| UWave All | 95.8 | 99.9 | 100.0 | 0.80 | 6.91 | 7.37 |
| Synt. Control | 99.4 | 100.0 | 100.0 | 0.90 | 2.84 | 2.78 |
| uWave-X | 91.3 | 99.6 | 99.6 | 1.34 | 5.47 | 5.17 |
| uWave-Z | 85.8 | 99.9 | 99.9 | 1.33 | 7.06 | 7.12 |
| Wafer | 99.8 | 100.0 | 100.0 | 0.88 | 1.93 | 1.94 |
| Yoga | 72.4 | 99.9 | 99.8 | 1.04 | 1.98 | 1.96 |
| Overall | 85.9 | 98.7 | **98.0** | **1.09** | 4.32 | 4.28 |

slightly lower. Importantly, TSCP tends to produce smaller confidence sets while still ensuring high coverage, which aligns more closely with our overall goal.

## 6.3 DOMAIN GENERALIZATION IN VEHICLE SENSOR DATA

Here, we conducted a comparative analysis of CP, RSCP and TSCP using our internal dataset derived from vehicle sensor data. In this context, we train the time series transformer using portions of data from each distinct domain, each characterized by unique recording configurations. Both the calibration and test sets consist of one or more configurations, each containing a minimum of 2 000 data points. The training set encompasses the rest, amounting to a total of 32 000 data points.

In Figure 3, we plot the accuracy, coverage and set-size for each specific test domain (configuration). In this context, we consider a temporal-warping transformation $(\sigma = 0.2)$ for TSCP and jitter (Gaussian noise with $\sigma = 0.2$) for

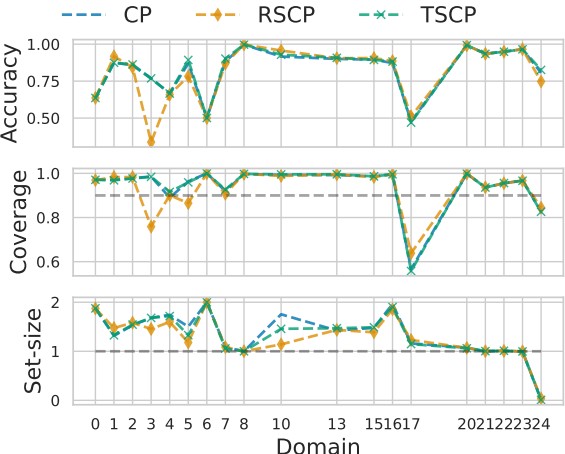

Figure 3: Coverage comparison between temporal smooth and plain CP in a domain generalization context.

RSCP. We emphasize, with a dashed gray line, the goal of achieving target coverage at 0.9 ($\alpha = 0.1$) and a target set size of 1. We observe that CP and TSCP demonstrate similar performance, whereas RSCP lags in certain configurations. Overall, for CP and TSCP, only 2 out of 19 configurations fall below the 0.9 threshold. This indicates that the target coverage for domain generalization is nearing its optimal level ($2/19 \simeq 0.1$).

In Table 3, we provide an average of the domain generalization performance between CP, RSCP and TSCP across different transformations. Interestingly, the coverage level is fairly uniform across these different transformations, illustrating the classifiers' resilience in preserving their prediction accuracy despite the diversity introduced by these transformations. However, to some degree, the performance in accuracy varies among transformations, with window-warp showing a significant decrease. This suggests that certain transformations can introduce complexities that challenge the classifier's ability to generalize. Regarding set-size, which reflects consistent outcomes, RSCP displays a surprisingly smaller value, yet it is in line with the results of the other methods. With these results, we want to highlight the effects of induced transformations on classifier performance, showing that while some transformations can enhance generalization, others may introduce challenges, impacting accuracy and certainty in predictions.

### 6.4 DISCUSSION OF RESULTS

In our analysis, spanning adversarial robustness and domain generalization, the study reveals notable insights into the performance of conformal methods such as CP, RSCP, and TSCP. Particularly in adversarial settings, the resilience of RSCP and TSCP shows that these methods maintain higher coverage against increasing adversarial perturbations, a practical demonstration of their robustness. This effect is offset

Table 3: Comparison of domain generalization performance between vanilla and $\pi$-smoothed classifiers across various transformations ($\sigma = 0.2$) in terms of accuracy (top-1), coverage and set-size. The values presented are averages calculated over the plain test sets of each individual configuration.

| Method | Tranform. | Acc. | Coverage | Set-Size |
|--------|-----------|------|----------|----------|
| CP | Vanilla | 83.0 | 94.2 | 1.35 |
| RSCP | Jitter | 80.5 | 93.0 | 1.28 |
| TSCP | Scaling | 83.1 | 93.8 | 1.33 |
| TSCP | Magnitude-Warp | 73.2 | 91.3 | 1.33 |
| TSCP | Time-Warp | 74.3 | 91.9 | 1.32 |
| TSCP | Window-Warp | 62.5 | 85.6 | 1.37 |

by a modest rise in set-size, hinting at reduced precision, yet it still preserves both accuracy and coverage effectively. TSCP generally performs well, often showing good accuracy and coverage. Furthermore, in the real-world application of vehicle sensor data, the experiments demonstrate that temporal transformations are nearing optimal target coverage for domain generalization, though the accuracy decreases with stronger transformations such as window-warping, pointing to challenges in this area. These findings collectively demonstrate the potential of introducing native time series augmentation in environments susceptible to domain shifts and highlight the challenges in enhancing classifier robustness and accuracy across diverse configurations.

## 7 CONCLUSION

In this work, we introduce a generalized smoothed classifier, inspired by the limitations of traditional RS under substantial perturbations like time-warping. By extending RS to include arbitrary perturbations specific to time series data and integrating it with CP, we have developed a robust approach that adapts to diverse and unknown distributions. We establish robustness boundaries for transformations in time series data, applying these to delineate the limits of confidence sets and to facilitate the transfer of learning across various domains. Additionally, we sketch a method for empirically estimating PAC guarantees in the context of domain generalization.

In a practical application, we present TSCP, a conformal prediction model that employs temporal-shift transformations to refine the input data. TSCP is designed to provide resilience against adversarial attacks and to adapt effectively to real-world data augmentations. The results highlight TSCP's effectiveness in domain generalization and empirical risk minimization, showcasing its practical utility. Overall, our work not only presents a theoretical advancement in handling perturbations in time series data but also demonstrates tangible benefits in real-world scenarios, bridging the gap between theoretical robustness and practical applicability.

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

# A   REAL-WORLD PERTURBATIONS FOR TIME SERIES DATA

Here, we introduce five primary techniques to augment time series: *jitter*, *scaling*, *magnitude warping*, *time warping* and *window warping*. Each method provides a unique way of altering the amplitude and frequency of time series data.

**Jitter**   The process of jittering, which involves introducing Gaussian noise to time series data, stands as a straightforward yet powerful method of transformation-based data augmentation [Iwana and Uchida, 2021a]. This operation can be mathematically represented as:

$$\tilde{x} = x_1 + \delta_1, \ldots, x_t + \delta_t, \ldots, x_T + \delta_T, \tag{30}$$

where each time step $t$ sees Gaussian noise $\delta_i$ added, following a normal distribution $\delta_i \sim \mathcal{N}(0, \sigma^2)$. The standard deviation $\sigma > 0$ of this noise is a tunable hyperparameter.

**Scaling**   Scaling pertains to modifying the global magnitude or intensity of a time series through the multiplication of a random scalar value. With a scaling parameter denoted as $\delta$, the scaling process is expressed as:

$$\tilde{x} = \delta x_1, \ldots, \delta x_t, \ldots, \delta x_T. \tag{31}$$

The scaling parameter $\delta$ can be derived from a Gaussian distribution $\delta \sim \mathcal{N}(1, \sigma^2)$ with $\sigma$ as a tunable hyperparameter, or it could be selected as a random value from a predetermined set. In the context of time series, the term scaling carries different meanings with respect to image processing, where it is associated to contrast. In time series, scaling strictly refers to the amplification of the magnitude of the elements, without altering the duration of the time series.

**Magnitude warping**   Magnitude warping, as presented by Um et al. [2017], is a data augmentation strategy specifically designed for time series data. This technique alters the magnitude of a signal through the application of a smoothed curve. Formally, the newly generated, or augmented, time series is derived through the following expression:

$$\tilde{x} = \delta_1 x_1, \ldots, \delta_t x_t, \ldots, \delta_T x_T, \tag{32}$$

where $\delta_1, \ldots, \delta_t, \ldots, \delta_T$ is a sequence derived from interpolating a cubic spline $S(u)$ with knots $u = u_1, \ldots, u_i, \ldots, u_I$. Each knot $u_i$ originates from a distribution $\mathcal{N}(1, \sigma^2)$, with the number of knots $I$ and the standard deviation $\sigma$ acting as tunable hyperparameters. The core concept of magnitude warping is to introduce minor variations in the data by either amplifying or diminishing random segments of the time series.

**Time warping**   Time-warping involves stretching or compressing the time axis to induce variability in the temporal dynamics. Given a univariate time series $x \in \mathbb{R}^T$ subjected to a time-warping perturbation characterized by a smooth warping path, the resulting augmented time series can be denoted as:

$$\tilde{x} = x_{\phi(1)}, \ldots, x_{\phi(t)}, \ldots, x_{\phi(T)}. \tag{33}$$

In this representation, $\phi(\cdot)$ is a time-warping function, which modifies the time indices based on a smooth curve. In previous works [Le Guennec et al., 2016, Um et al., 2017, Iwana and Uchida, 2021b], this curve was characterized by a cubic spline, $S(u)$, having knots defined as $u = u_1, \ldots, u_i, \ldots, u_I$, where each knot height, $u_i$, was derived from a normal distribution, $u_i \sim \mathcal{N}(1, \sigma^2)$. However, in this work we consider a different approach to temporally shift the time series.

**Window warping**   A familiar technique of time warping termed as window warping has been introduced by Le Guennec et al. [2016]. In this method, a random segment of the time series, starting from $p \in \mathbb{N}$ with $0 < p < T$ and ending at $p + \lceil \sigma \cdot T \rceil$, is selected and either stretched by a factor of 2 or contracted by a factor of $\frac{1}{2}$. Then the segment is interpolated back into the original time series. Even though the stretching and contracting factors are preset to 2 and $\frac{1}{2}$ respectively, these values can be adjusted or optimized to other values as needed.

# B   PROOF OF THEOREM 4.1

**Theorem 4.1.** *Assume a set of samples $\{(x^{(i)}, y^{(i)})\}_{i=1}^{n+1}$ that are exchangeably drawn from an unknown distribution $\mathcal{D}_{xy}$. Let $\phi : \mathcal{X} \times \mathcal{Z} \to \mathcal{X}$ be a differentially resolvable transformation, let $\mathcal{Z}_\lambda \subseteq \mathcal{Z}$, $\{\lambda_j\}_{j=1}^N$ be a set of perturbation parameters*

and let $G : \mathcal{X} \to \mathbb{P}(\mathcal{Y})$ be a smooth classifier as in Def. 3.2 that predicts $y_A \in \mathcal{Y}$ given $x$ (i.e. $G(y_A \mid x)$ where $x = x^{(n+1)}$). If for any $j$, $G(x)$ has class probabilities that satisfy:

$$G(y_A \mid \phi(x, \lambda_j)) \geq p_A^{(j)} \geq p_B^{(j)} \geq \max_{y \neq y_A} G(y \mid \phi(x, \lambda_j)), \tag{16}$$

and Eq. 14 holds, then, the prediction set $\mathcal{C}_\pi$ as defined in Eq. 15 will satisfy the following probability:

$$\mathbb{P}[y^{(n+1)} \in \mathcal{C}_\pi(\phi(x^{(n+1)}, \pi))] \geq 1 - \alpha. \tag{17}$$

*Proof.* From Corollary 2 of Li et al. [2021], we know that if the maximum interpolation error satisfy Eq. 14, then it is guaranteed that $\forall \lambda \in \mathcal{Z}_\lambda : y_A = \arg\max_y G_\phi(y \mid \phi(x, \lambda)$. Therefore, if we define the robustness certificates radius as:

$$R_\pi \overset{\text{def}}{=} \frac{\sigma}{2} \min_{1 \leq j \leq N} \left( \phi^{-1}(p_A^{(j)}) - \phi^{-1}(p_B^{(j)}) \right),$$

we can link the observed score $S_\phi(\tilde{x}^{(n+1)}, y)$ with the unobserved score $S_\phi(x^{(n+1)}, y)$ for any given $y \in \mathcal{Y}$. Thus, let us consider the definition of the conformal set as in Eq. 15:

$$\mathbb{P}\left[ y^{(n+1)} \in \mathcal{C}_\pi(\tilde{x}^{(n+1)}) \right] = \mathbb{P}\left[ S_\phi(\tilde{x}^{(n+1)}, y^{(n+1)}) \leq Q_{1-\alpha}(\{S^{(i)}\}_{i \in \mathcal{D}_{cal}}) + R_\pi \right]$$

$$\text{(Eq. 12)} \quad \geq \mathbb{P}\left[ S_\phi(x^{(n+1)}, y^{(n+1)}) + R_\pi \leq Q_{1-\alpha}(\{S^{(i)}\}_{i \in \mathcal{D}_{cal}}) + R_\pi \right]$$

$$= \left[ S_\phi(x^{(n+1)}, y^{(n+1)}) \leq Q_{1-\alpha}(\{S^{(i)}\}_{i \in \mathcal{D}_{cal}}) \right]$$

$$\text{(Eq. 1)} \quad \geq 1 - \alpha$$

$\square$

## C   LIPSCHITZ CONSTANT FOR CUBIC SPLINE INTERPOLATION

When dealing with magnitude and window warping, estimating the maximum error in interpolation involves finding the largest value of the derivative of the cubic spline used for interpolation. This maximum value is considered as the Lipschitz constant. The cubic spline is a piecewise polynomial function, typically of degree three. Assume we have a sequence of $n + 1$ knots, $(x_0, y_0)$ through $(x_n, y_n)$. There exists a cubic spline segment $q_i(x)$ defined as:

$$q_i(x) = (1 - t(x))y_{i-1} + t(x)y_i + t(x)(1 - t(x))((1 - t(x))a_i + t(x)b_i),$$
$$\text{with} \quad t(x) = \frac{x - x_{i-1}}{x_i - x_{i-1}}, \quad a_i = k_{i-1}(x_i - x_{i-1}) - (y_i - y_{i-1}), \quad b_i = -k_i(x_i - x_{i-1}) + (y_i - y_{i-1}), \tag{34}$$

where $k_i$ represents the second order derivative of the spline at the knot points $(x_i, y_i)$. To compute the derivative of the cubic spline function $q_i(x)$, we first need to recognize that $q_i(x)$ is a composite function involving $t$ which itself is a function of $x$. Therefore, we will use the chain rule to find the derivative, i.e. $\frac{dq_i}{dx} = \frac{dq_i}{dt} \cdot \frac{dt}{dx}$. Thus, the first order derivative is defined as:

$$\frac{dq_i}{dx} = \frac{y_i - y_{i-1}}{x_i - x_{i-1}} + (1 - 2t)\frac{a_i(1 - t) + b_i t}{x_i - x_{i-1}} + t(1 - t)\frac{b_i - a_i}{x_i - x_{i-1}}, \tag{35}$$

where we omit the dependence of $t$ on $x$ for brevity. This derivative represents the rate of change of the cubic spline segment $q_i(x)$ with respect to $x$, and it varies along different segments of the spline depending on the values of $x_i, x_{i-1}, y_i, y_{i-1}, k_i$, and $k_{i-1}$. From the spline's derivative, the Lipschitz constant can be estimated by finding the maximum of its absolute values. The maximum value of the first derivative occurs either at the endpoints of a segment (i.e., at the knots $x_{i-1}$ or $x_i$) or at a critical point within the segment where the second-order derivative is zero. The second order derivative $\frac{d^2 q_i}{dx^2}$ gives the maximum rate of change of the first derivative. Thus, let us compute $\frac{d^2 q_i}{dx^2}$ and set it equal to zero. The second derivative of the cubic spline function $q_i(x)$ is:

$$\frac{d^2 q_i}{dx^2} = 2\frac{b_i - 2a_i + (a_i - b_i)3t}{(x_i - x_{i-1})^2}. \tag{36}$$

Next, we set this second derivative to zero and solve for $x$. This will give us the points where the curvature of the spline segment changes, indicating inflection points. The solution to the equation $\frac{d^2 q_i}{dx^2} = 0$ is:

$$t = \frac{2a_i - b_i}{3(a_i - b_i)}, \quad \text{or} \quad x = \frac{(2a_i - b_i)x_i + (a_i - 2b_i)x_{i-1}}{3(a_i - b_i)}. \tag{37}$$

This formula represents the inflection point of the spline segment between $x_{i-1}$ and $x_i$. Inflection points are where the curvature of the spline changes sign and we can obtain the maximum value of the first order derivative by inserting $t$ (or $x$) in $\frac{dq_i}{dx}$. In practice, the specific value where this occurs depend on the values of $x_i$, $x_{i-1}$, $y_i$, $y_{i-1}$, $k_i$, and $k_{i-1}$. The global maximum of the first derivative of the entire cubic spline is the largest value found among all segments.

## D  ADDITIONAL DETAILS ON EXPERIMENTAL PROCEDURES

In Table 4, we provide an overview of the architecture and key features of two neural network models: a Convolutional Neural Network (CNN) and a Time Series Transformer. The CNN comprises three convolutional layers with 32, 64, and 64 channels respectively, and two linear layers with 128 and 32 units. It includes max pooling with a kernel size of 4 and flattening operations. The Time Series Transformer, in contrast, does not have convolutional layers but includes two transformer layers and two linear layers, each with 32 units, along with a flattening step. Both models utilize ReLU activation functions in their convolutional and linear layers, and they both have a softmax output activation function.

| Parameter | CNN | Time Series Transformer |
|---|---|---|
| Number of Layers | 3 Conv + 2 Linear | 1 Transformer + 2 Linear |
| Convolutional Layers | 3 (32, 64, 64 channels) | N/A |
| Transformer Layers | N/A | 2 Layers |
| Max Pooling | Yes (Kernel Size: 4) | N/A |
| Linear Layers | 2 (128, 32 units) | 2 (32 units each) |
| Activation Functions | ReLU | ReLU |
| Output Activation | Softmax | Softmax |

Table 4: Network parameters of CNN and time series transformer networks.

In training the respective networks, both the CNN and the transformer shared similar hyper-parameters. Both models were training for 200 epochs with a batch size of 1024 and implementing an early stopping mechanism with a patience of 100 epochs to prevent overfitting. We incorporate random data augmentation from App. A with an intensity of $0.5$. We utilize the Adam optimizer for both models and paired with a learning rate adjustment strategy that reduces the rate upon hitting a plateau.

## E  ADVERSARIAL ATTACK EXPERIMENT DETAILS

An adversarial attack refers to crafting input data with the intent of fooling a machine learning model into making a misclassification [Szegedy et al., 2013, Goodfellow et al., 2014]. Formally, $\tilde{x}$ is called an adversarial example of $x$ if $\arg\max_{y \in \mathcal{Y}} F_y d(\tilde{x}) \neq \arg\max_{y \in \mathcal{Y}} F_y(x)$ where $d(\tilde{x}, x) \leq \epsilon$, with $\epsilon > 0$. In practice, using a loss function $\mathcal{L}$ as defined in Carlini and Wagner [2017]:

$$\mathcal{L}^{target}(\tilde{x}) = \max_{\tilde{y} \in \mathcal{Y} \backslash y} F_{\tilde{y}}(\tilde{x}) - F_y(\tilde{x}), \tag{38}$$

the goal is to maximize this difference in order to make the model very confident about the wrong classification.

We report in Table 5, the clean top-1 accuracy, the adversarial accuracy, coverage and set-size for CP, RSCP and TSCP under an uniform distribution of 20 adversarial attacks within $\epsilon \in [0, 0.1]$. In this comparison, we consider only the results were the classifier achieved a clean accuracy higher than 70%.

TSCP consistently outshines CP and RSCP, particularly in maintaining higher adversarial accuracy and coverage, demonstrating its superior resilience to adversarial manipulations. This robustness is evident despite the noticeable decline in performance all methods experience under adversarial conditions compared to their clean top-1 accuracy. The performance

of these methods, however, varies significantly across different datasets, underscoring the influence of dataset characteristics on model robustness. For instance, in datasets like *ECG200* and *Plane*, all methods maintain high adversarial accuracy, whereas in others like *Meat* and *Coffee*, there's a substantial performance drop, especially for CP and RSCP. Furthermore, TSCP tends to generate more precise predictions, as indicated by its generally smaller set sizes compared to RSCP, while CP, though having the smallest set sizes, lags in adversarial accuracy.

## F  DOMAIN GENERALIZATION FOR TIME SERIES CLASSIFICATION

In Figure 4, we display the individual results of the domain generalization experiment conducted in the in-house dataset. Each test set satisfies the condition of a minimum of 2 000 samples. The ability of the model to maintain high accuracy and coverage, along with a consistent set size across these different domains, is indicative of its robustness and effectiveness in handling unseen domains. As we observe, the use of jitter transformation to augment the input signal shows reduced coverage performance in two additional domains compared to other transformations.

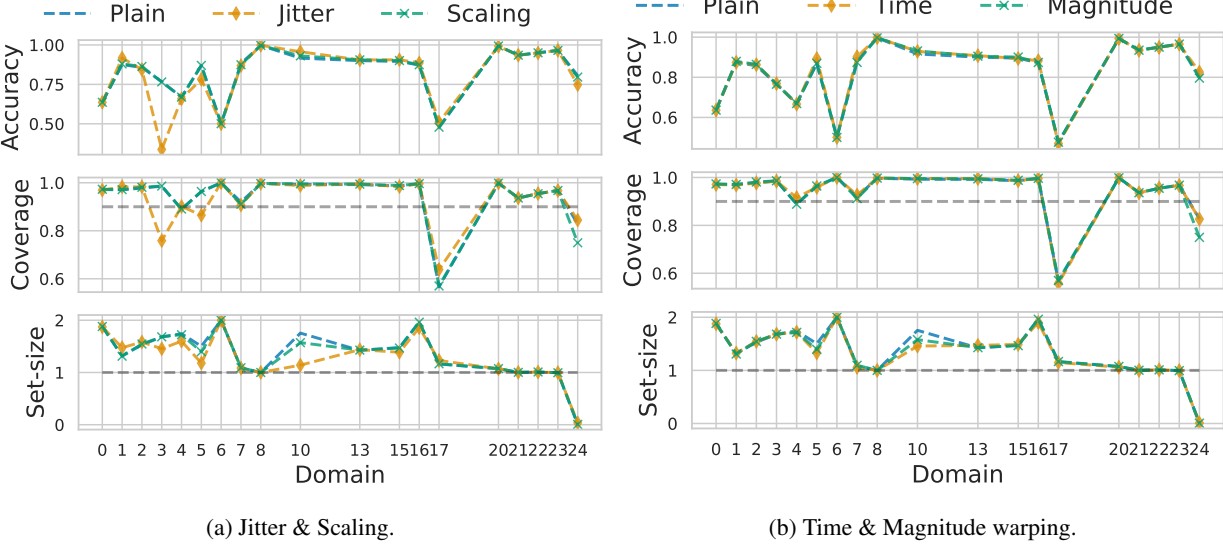

(a) Jitter & Scaling.
(b) Time & Magnitude warping.

Figure 4: Results of domain generalization for a binary time-series classifier applied to different recordings of vehicle sensor data.

Table 5: Comparison analysis of CP [Vovk et al., 2005], RSCP [Gendler et al., 2021] and TSCP (our) across UCR [Dau et al., 2018] datasets. We consider an average of 20 uniformly distributed PGD [Carlini and Wagner, 2017] attack samples with $\epsilon \in [0, 0.1]$ and a target coverage of 90% ($\alpha = 0.1$). RSCP and TSCP are augmented by $\sigma = 0.2$. Continue in Table 6.

| Dataset | Acc. | Adversarial Acc. | | | Coverage | | | Set-Size | | |
|---|---|---|---|---|---|---|---|---|---|---|
| | | CP | RSCP | TSCP | CP | RSCP | TSCP | CP | RSCP | TSCP |
| ArrowHead | 72.0 | 57.8 | 58.0 | 60.7 | 78.7 | 97.5 | 99.4 | 0.83 | 2.41 | 2.53 |
| BME | 93.3 | 90.4 | 90.4 | 90.4 | 100.0 | 100.0 | 100.0 | 0.47 | 3.00 | 3.00 |
| Beef | 70.0 | 51.6 | 50.0 | 49.0 | 71.6 | 100.0 | 100.0 | 0.40 | 4.57 | 4.57 |
| BirdChicken | 75.0 | 72.1 | 72.1 | 72.1 | 78.2 | 100.0 | 100.0 | 0.74 | 2.00 | 2.00 |
| CBF | 92.6 | 91.3 | 91.3 | 91.3 | 97.2 | 100.0 | 100.0 | 2.14 | 3.00 | 3.00 |
| Car | 73.3 | 59.5 | 59.5 | 59.5 | 78.4 | 100.0 | 100.0 | 0.95 | 4.00 | 4.00 |
| Chinatown | 87.5 | 87.5 | 87.5 | 87.5 | 92.2 | 100.0 | 100.0 | 0.69 | 2.00 | 2.00 |
| CinC-ECG-torso | 73.9 | 69.9 | 69.4 | 68.8 | 88.6 | 99.9 | 98.2 | 0.52 | 3.98 | 3.84 |
| Coffee | 96.4 | 54.8 | 55.4 | 53.9 | 56.4 | 99.1 | 53.9 | 0.61 | 1.94 | 0.99 |
| Cricket-X | 70.5 | 65.6 | 66.8 | 62.5 | 85.4 | 99.7 | 100.0 | 1.22 | 10.54 | 10.19 |
| Cricket-Z | 70.3 | 65.2 | 66.1 | 62.5 | 85.0 | 99.4 | 99.7 | 1.94 | 10.20 | 9.70 |
| DiatomSizeReduction | 94.1 | 68.3 | 68.3 | 33.1 | 75.0 | 99.9 | 99.3 | 0.69 | 3.99 | 3.94 |
| DistalPhalanxOutlineAgeGroup | 85.2 | 79.1 | 78.9 | 81.0 | 92.1 | 100.0 | 100.0 | 1.57 | 2.93 | 2.96 |
| DistalPhalanxOutlineCorrect | 77.7 | 66.2 | 61.2 | 46.5 | 93.3 | 99.8 | 100.0 | 1.72 | 1.97 | 1.97 |
| DistalPhalanxTW | 74.8 | 72.4 | 73.4 | 76.2 | 90.6 | 100.0 | 100.0 | 2.17 | 4.48 | 4.35 |
| ECG200 | 89.0 | 86.4 | 86.7 | 84.6 | 92.6 | 100.0 | 100.0 | 0.79 | 1.98 | 1.94 |
| ECG5000 | 93.6 | 92.2 | 91.4 | 92.4 | 96.0 | 99.4 | 99.8 | 0.93 | 4.01 | 4.46 |
| ECGFiveDays | 80.8 | 72.9 | 72.3 | 83.7 | 78.8 | 97.8 | 93.0 | 0.77 | 1.59 | 1.25 |
| Earthquakes | 73.3 | 71.1 | 69.8 | 74.8 | 82.9 | 94.9 | 92.8 | 0.39 | 1.64 | 1.42 |
| ElectricDevices | 73.8 | 64.8 | 66.7 | 64.6 | 84.0 | 98.1 | 98.1 | 1.63 | 6.37 | 6.44 |
| FaceAll | 72.7 | 67.9 | 67.9 | 72.0 | 90.7 | 87.0 | 88.5 | 0.92 | 8.82 | 7.60 |
| FaceFour | 79.5 | 69.4 | 68.2 | 81.1 | 88.1 | 95.9 | 99.3 | 1.69 | 2.38 | 2.30 |
| FacesUCR | 83.1 | 73.2 | 73.7 | 81.3 | 91.4 | 99.5 | 99.5 | 1.04 | 11.24 | 11.64 |
| Fish | 86.9 | 48.7 | 48.7 | 48.7 | 63.8 | 100.0 | 100.0 | 0.70 | 7.00 | 7.00 |
| FordA | 90.1 | 70.9 | 72.2 | 78.3 | 75.9 | 95.8 | 97.4 | 0.83 | 1.66 | 1.66 |
| FordB | 85.2 | 72.5 | 72.0 | 76.4 | 78.0 | 99.6 | 98.2 | 1.04 | 1.73 | 1.66 |
| FreezerRegularTrain | 96.1 | 76.3 | 76.3 | 76.3 | 72.9 | 100.0 | 100.0 | 0.34 | 2.00 | 2.00 |
| FreezerSmallTrain | 73.3 | 73.0 | 73.0 | 73.0 | 89.3 | 100.0 | 100.0 | 0.32 | 2.00 | 2.00 |
| GunPointAgeSpan | 87.3 | 87.3 | 87.3 | 87.3 | 99.3 | 100.0 | 100.0 | 1.63 | 2.00 | 2.00 |
| GunPointMaleVersusFemale | 93.4 | 93.1 | 93.1 | 93.1 | 99.6 | 100.0 | 100.0 | 1.11 | 2.00 | 2.00 |
| GunPointOldVersusYoung | 89.2 | 89.0 | 89.0 | 89.0 | 91.9 | 100.0 | 100.0 | 0.91 | 2.00 | 2.00 |
| Gun-Point | 97.3 | 71.1 | 70.8 | 82.4 | 87.9 | 100.0 | 100.0 | 0.48 | 1.97 | 1.96 |
| HandOutlines | 84.7 | 59.5 | 62.4 | 78.3 | 59.2 | 98.7 | 98.5 | 0.89 | 1.94 | 1.92 |
| HouseTwenty | 72.3 | 72.3 | 72.3 | 72.3 | 79.7 | 100.0 | 100.0 | 0.62 | 2.00 | 2.00 |
| InsectEPGRegularTrain | 100.0 | 100.0 | 100.0 | 100.0 | 100.0 | 100.0 | 100.0 | 0.65 | 3.00 | 3.00 |
| InsectEPGSmallTrain | 100.0 | 100.0 | 100.0 | 100.0 | 100.0 | 100.0 | 100.0 | 0.51 | 3.00 | 3.00 |
| ItalyPowerDemand | 95.8 | 90.0 | 90.0 | 90.0 | 98.1 | 100.0 | 100.0 | 0.70 | 2.00 | 2.00 |
| LargeKitchenAppliances | 72.0 | 60.6 | 62.3 | 66.2 | 84.1 | 98.1 | 97.3 | 1.64 | 2.80 | 2.70 |
| Lighting2 | 75.4 | 72.8 | 72.4 | 72.3 | 82.4 | 100.0 | 100.0 | 0.48 | 1.96 | 1.91 |
| Lighting7 | 71.2 | 65.6 | 64.4 | 60.0 | 85.1 | 100.0 | 100.0 | 1.50 | 6.31 | 5.41 |
| MALLAT | 94.7 | 78.4 | 79.8 | 85.8 | 78.7 | 100.0 | 100.0 | 0.49 | 6.81 | 6.89 |
| Meat | 85.0 | 20.2 | 21.3 | 26.5 | 23.9 | 62.0 | 61.1 | 0.84 | 1.97 | 1.91 |
| MiddlePhalanxOutlineAgeGroup | 79.8 | 75.8 | 75.4 | 76.3 | 95.1 | 89.5 | 94.2 | 1.70 | 1.45 | 2.01 |
| MiddlePhalanxOutlineCorrect | 70.8 | 58.6 | 58.5 | 58.6 | 91.9 | 100.0 | 100.0 | 1.72 | 2.00 | 2.00 |
| MixedShapesRegularTrain | 88.9 | 83.3 | 83.3 | 83.3 | 93.8 | 100.0 | 100.0 | 1.08 | 5.00 | 5.00 |
| MixedShapesSmallTrain | 80.8 | 76.3 | 76.3 | 76.3 | 90.8 | 100.0 | 100.0 | 1.32 | 5.00 | 5.00 |
| MoteStrain | 77.5 | 77.5 | 78.0 | 79.2 | 87.7 | 98.2 | 94.2 | 0.49 | 1.73 | 1.35 |
| ⋮ | ⋮ | ⋮ | ⋮ | ⋮ | ⋮ | ⋮ | ⋮ | ⋮ | ⋮ | ⋮ |

Table 6: Continuation of Table 5

| Dataset | Acc. | Adversarial Acc. | | | Coverage | | | Set-Size | | |
|---|---|---|---|---|---|---|---|---|---|---|
| | | CP | RSCP | TSCP | CP | RSCP | TSCP | CP | RSCP | TSCP |
| | ⋮ | ⋮ | ⋮ | ⋮ | ⋮ | ⋮ | ⋮ | ⋮ | ⋮ | ⋮ |
| NonInvasiveFatalECG-Thorax1 | 83.3 | 33.9 | 35.4 | 38.5 | 70.7 | 100.0 | 100.0 | 3.21 | 32.22 | 33.91 |
| NonInvasiveFatalECG-Thorax2 | 89.5 | 37.2 | 38.7 | 45.9 | 65.1 | 99.9 | 100.0 | 2.20 | 27.47 | 27.23 |
| Plane | 96.2 | 95.6 | 95.4 | 91.0 | 99.2 | 100.0 | 100.0 | 1.06 | 4.93 | 4.78 |
| PowerCons | 98.3 | 95.3 | 95.3 | 95.3 | 100.0 | 100.0 | 100.0 | 0.75 | 2.00 | 2.00 |
| ProximalPhalanxOutlineAgeGroup | 83.4 | 73.8 | 75.5 | 81.0 | 80.3 | 100.0 | 100.0 | 1.15 | 2.99 | 2.99 |
| ProximalPhalanxOutlineCorrect | 75.9 | 66.7 | 74.8 | 72.3 | 91.2 | 100.0 | 100.0 | 1.55 | 1.98 | 1.96 |
| ProximalPhalanxTW | 73.0 | 70.8 | 68.8 | 65.3 | 88.7 | 100.0 | 100.0 | 2.11 | 4.56 | 5.25 |
| Rock | 74.0 | 75.4 | 75.4 | 75.4 | 100.0 | 100.0 | 100.0 | 3.42 | 4.00 | 4.00 |
| SemgHandGenderCh2 | 86.8 | 86.1 | 86.1 | 86.1 | 93.4 | 100.0 | 100.0 | 0.66 | 2.00 | 2.00 |
| SemgHandSubjectCh2 | 76.2 | 75.8 | 75.8 | 75.8 | 89.2 | 100.0 | 100.0 | 1.30 | 5.00 | 5.00 |
| SonyAIBORobotSurface1 | 83.7 | 81.5 | 81.5 | 81.5 | 89.4 | 100.0 | 100.0 | 0.73 | 2.00 | 2.00 |
| SonyAIBORobotSurface2 | 81.1 | 82.0 | 82.0 | 82.0 | 92.7 | 100.0 | 100.0 | 0.68 | 2.00 | 2.00 |
| StarLightCurves | 92.0 | 83.7 | 84.1 | 86.4 | 95.8 | 100.0 | 100.0 | 1.06 | 2.88 | 2.85 |
| Strawberry | 86.9 | 53.5 | 55.5 | 64.4 | 58.7 | 100.0 | 100.0 | 1.11 | 1.95 | 1.91 |
| SwedishLeaf | 85.6 | 59.8 | 63.3 | 57.6 | 82.4 | 100.0 | 100.0 | 1.64 | 13.96 | 13.66 |
| Symbols | 82.0 | 76.7 | 76.4 | 83.2 | 88.1 | 99.9 | 99.8 | 0.95 | 4.77 | 4.04 |
| ToeSegmentation1 | 77.2 | 75.7 | 75.7 | 77.2 | 79.2 | 99.4 | 95.4 | 0.89 | 1.81 | 1.64 |
| ToeSegmentation2 | 86.2 | 84.0 | 84.1 | 87.3 | 93.9 | 98.7 | 95.6 | 0.66 | 1.45 | 1.38 |
| Trace | 99.0 | 82.4 | 81.9 | 83.1 | 90.1 | 100.0 | 100.0 | 0.68 | 3.14 | 3.18 |
| TwoLeadECG | 89.5 | 65.4 | 66.0 | 64.5 | 75.0 | 95.4 | 99.7 | 0.44 | 1.81 | 1.92 |
| Two-Patterns | 99.9 | 99.9 | 100.0 | 99.9 | 100.0 | 100.0 | 100.0 | 0.78 | 3.80 | 3.77 |
| UMD | 91.7 | 92.8 | 92.8 | 92.8 | 99.7 | 100.0 | 100.0 | 1.71 | 3.00 | 3.00 |
| UWaveGestureLibraryAll | 94.7 | 87.9 | 89.0 | 93.0 | 95.8 | 99.9 | 100.0 | 0.80 | 6.91 | 7.37 |
| synthetic-control | 97.3 | 95.5 | 95.0 | 97.6 | 99.4 | 100.0 | 100.0 | 0.90 | 2.84 | 2.78 |
| uWaveGestureLibrary-X | 80.2 | 75.8 | 75.1 | 75.6 | 91.3 | 99.6 | 99.6 | 1.34 | 5.47 | 5.17 |
| uWaveGestureLibrary-Z | 70.3 | 65.2 | 65.3 | 69.3 | 85.8 | 99.9 | 99.9 | 1.33 | 7.06 | 7.12 |
| wafer | 98.9 | 98.7 | 98.7 | 97.2 | 99.8 | 100.0 | 100.0 | 0.88 | 1.93 | 1.94 |
| yoga | 75.3 | 67.5 | 67.3 | 67.4 | 72.4 | 99.9 | 99.8 | 1.04 | 1.98 | 1.96 |
| Overall | 84.1 | 74.1 | 74.4 | **75.3** | 85.9 | 98.7 | **98.0** | **1.09** | 4.32 | 4.28 |