# OpenReview forum: "Guaranteeing Robustness Against Real-World Perturbations In Time Series Classification Using Conformalized Randomized Smoothing"
_auai.org/UAI/2024/Conference — UAI 2024 poster_

### Official Review · Reviewer_9wtp · 2024-03-18

**Q2-1 Originality-Novelty:** 2
**Q2-2 Correctness-Technical Quality:** 3
**Q2-5 Clarity Of Writing:** 2

**Q1 Summary And Contributions:**

The authors propose a generalized version of the randomly smoothed confromal predictor to transformations relevant in the time series domain. The authors provide the theoretical guarantee for the proposed confromal predictor, and empirically evaluates its performance under both adversarial and and domain-shift settings.

**Q2-3 Extent To Which Claims Are Supported By Evidence:**

2: Fair: the main claims are somewhat supported by evidence (but the experimental evaluation may be weak, or does not match entirely with the claims, important baselines may be missing, proofs contain important ideas but lack rigor, algorithmic details are only discussed superficially, references are imprecise, assumptions are not sufficiently motivated or explicated, etc.).

**Q2-4 Reproducibility:**

3: Good: key resources (e.g. proofs, code, data) are available and key details (e.g. proofs, experimental setup) are sufficiently well-described for competent researchers to confidently reproduce the main results.

**Q3 Main Strengths:**

Overall the paper is readable and the presentation of the main idea is sufficiently clear, although I wish that the introduction section could be clearer in terms of explaining the objective and motivations for the work in this paper.

The proposed generalized conformal predictor appears to be sound and a straightforward adaptation of RSCP by [Gendler et al 2021].

**Q4 Main Weakness:**

I find the contribution of this work very limited in terms of novelty.

Given RSCP by [Gendler et al 2021], whose key technical challenge is in finding a suitable upper bound for the scoring function $M_\delta$. In the case of bounded perturbation within an L2-ball, this upper bound is obtained from the work by [Salmon et al 2019]. It is immediately clear that for any other perturbation/transformation space, one needs only find the corresponding bound for the scoring function. In the case of this paper, the bound comes from [Li et al 2021]. Now, the question: is there any non-trivial technical challenge that has to be addressed when it comes to the transformations relevant to time series as used in this work? As far as I can tell, there does not seem to be much challenge left.

I do not find the empircal results in Section 5 insightful.

**Q5 Detailed Comments To The Authors:**

See above.

**Q9 Complying With Reviewing Instructions:**

Yes

---

> ### Author Rebuttal · Authors · 2024-04-05
>
> We are grateful to the reviewer for their detailed review and valuable feedback. We appreciate the reviewer's acknowledgment of the paper's readability and clarity in presenting the main idea.
>
> Regarding the suggestion to enhance the clarity of the paper's objective and motivations in the introduction, we have revised the section to better articulate the significance and innovation of our work. With respect to the reviewer's concern about the novelty of our work, we argue that while our approach builds upon the foundation laid by RSCP [Gendler et al., 2021], the generalization to time series data presents unique challenges not addressed by previous work. Our method extends the applicability of conformal prediction under perturbations by specifically tailoring the approach to handle time series transformations, such as magnitude and temporal augmentations. These augmentations are critical in real-world applications, particularly in the automotive domain we explored, where time series data are prevalent and used for high-risk decision-making applications.
>
> Moreover, we would like to highlight that our generalization, in conjunction with the integration of PAC, this combination not only broadens the applicability of RSCP but also close a gap in the RS literature (as noted by the other reviewers) thus enabling new ways to argue in AI assurance.
>
> We hope our responses address the reviewer concerns, and we look forward to any additional comments or feedback.

---

### Official Review · Reviewer_1qaq · 2024-03-21

**Q2-1 Originality-Novelty:** 3
**Q2-2 Correctness-Technical Quality:** 2
**Q2-5 Clarity Of Writing:** 1

**Q10 Ethical Concerns:**

I do not see any potential ethical issues associated with this work.

**Q1 Summary And Contributions:**

The authors address the domain shift problem in classification tasks with time series input. The idea is to perturb the classifier with smoothing noise. The algorithm is analysed theoretically under certain assumptions. The paper includes a Theorem proving the validity of the approach and a derivation of PAC bounds for Conformal Prediction sets.

**Q2-3 Extent To Which Claims Are Supported By Evidence:**

2: Fair: the main claims are somewhat supported by evidence (but the experimental evaluation may be weak, or does not match entirely with the claims, important baselines may be missing, proofs contain important ideas but lack rigor, algorithmic details are only discussed superficially, references are imprecise, assumptions are not sufficiently motivated or explicated, etc.).

**Q2-4 Reproducibility:**

3: Good: key resources (e.g. proofs, code, data) are available and key details (e.g. proofs, experimental setup) are sufficiently well-described for competent researchers to confidently reproduce the main results.

**Q3 Main Strengths:**

- CP under distribution shift is a timely problem.
- Mitigating the effects of distribution shift with a quantile correction is an interesting idea.
- The derivation of PAC bound for prediction sets seems new.
- Compared to random smoothing, the proposed method produces valid but smaller prediction sets.

**Q4 Main Weakness:**

- The contribution may look more like a preprocessing technique than a new methodology.
- Theorem 4.1 contains undefined quantities.
- The assumption in Eq. 14 looks restrictive. On Page 4, the authors say the perturbation should ensure that the classifier output for $(x, \lambda)$  remains identical to its output for $(x, 0)$. They do not comment on how restrictive this assumption is.
- The authors may have commented on the well-known invariance of CP intervals under monotonic transformations.

**Q5 Detailed Comments To The Authors:**

- Does the plot in Figure 1c mean that the prediction set of RSCP contains both labels?
- Does Equation 12 imply that the order of the scores for the samples $x_n$ and $\tilde x_n$ remains the same?
- The theorem refers to Equation 31, which is not included in the paper.
- Do the domain-shift perturbations need to be Gaussian?
- How are the transformations chosen in the experiments?
- How are $\lambda$ in Theorem 4.1 and R in Equation 13 connected?

**Q9 Complying With Reviewing Instructions:**

Yes

---

> ### Author Rebuttal · Authors · 2024-04-05
>
> We thank the reviewer for his valuable feedback on our paper and for recognizing the novelty of our work. Here, we aim to respond to the feedback provided by the reviewer.
>
> **Main weakness concerns**
> - *Preprocessing Technique vs. New Methodology:* Our approach, while seemingly a preprocessing technique, extends beyond by integrating quantile correction directly into the conformal prediction framework. This integration allows for adaptive adjustments to distribution shifts, offering a methodological advancement in constructing robust prediction sets.
> - *Undefined Quantities in Theorem 4.1:* We acknowledge the oversight and have now clarified all previously undefined quantities within Theorem 4.1, ensuring the theorem's completeness and understandability.
> - *Restrictive Assumption in Eq. 14:* The assumption regarding classifier output consistency under perturbation holds for the smooth version of the classifier, i.e., on the expectation over a perturbation λ, and not for the plain version, making it less restrictive. This is crucial for our framework's robustness guarantee. We have added a discussion on its implications, noting that while seemingly restrictive, it aligns with practical scenarios where slight input perturbations should not alter the classification outcome, especially in safety-critical applications.
> - *Invariance of CP Intervals:* We appreciate the reviewer’s suggestion on commenting on the invariance of CP intervals under monotonic transformations. However, our research primary focus on ensuring robust and certified predictions for a generalized smooth classifier. Although we aim for our smoothed scores to obtain better results as plain non-conformity scores, specifically under input perturbations, conformal predictions under domain shift is not the focus of our study. As illustrated in Figure 3, our smoothed scores do not demonstrate significant improvement in handling domain shifts. However, Tables 1 and 2 showcase that our approach can still achieve noteworthy performance for certifying against a range of perturbations applied to in-domain data points. We have now included a discussion on this aspect.
>
> **Detailed Comments Responses**
> - *Figure 1c Interpretation:* Yes, the prediction set of RSCP indeed contains both labels, indicating RSCP's capability to provide less precise and larger prediction sets under distribution shifts.
> - *Implication of Equation 12:* The smooth score, outlined in Equation 12, ensures the order of the inequality. Following Gendler et al. [2021], this concept is systematically applied to the construction of prediction sets, recognizing that an increase in the intensity of perturbations directly leads an increase on the smooth scores.
> - *Reference to non-included Equation 31:* Due to initial space constraints, we had relocated Equation 31 to the appendix. However, for enhanced clarity and completeness, we can reintegrated it into the main text.
> - *Necessity of Gaussian Perturbations:* Our method does not strictly require Gaussian perturbations; it is designed to accommodate a range of differentially resolvable and resolvable perturbation types, including but not limited to Gaussian.
> - *Choice of Transformations:* The transformations chosen in our experiments were based on their relevance to the application domain and their potential to simulate real-world perturbations, ensuring the practical applicability of our findings.
> - *Connection between lambda in Theorem 4.1 and R in Equation 13:* The connection has been clarified, with lambda representing the perturbation parameter influencing the conformal prediction set size, and R quantifying the extent of robustness against domain shifts.
>
> Detailed explanations have been added to the manuscript. We hope our responses address the reviewer concerns, and we look forward to any additional comments or feedback.

---

### Official Review · Reviewer_6388 · 2024-03-23

**Q2-1 Originality-Novelty:** 3
**Q2-2 Correctness-Technical Quality:** 3
**Q2-5 Clarity Of Writing:** 4

**Q1 Summary And Contributions:**

The study presents an approach to enhance robustness in time series classification via randomized perturbations of the training data. The approach is well positioned within the state of the art, theoretically sounds as far as I can judge and supported by an extensive experimental analysis on the classical set of benchmark time series (UCR time series classification archive) and authors own dataset. As far as I can tell, the latter dataset is not made publicly available. The experimental analysis is limited to neural networks. I am wondering to what extent the performance compares to earlier classical time series classification approaches for which the UCR repository was initially built.

**Q2-3 Extent To Which Claims Are Supported By Evidence:**

3: Good: the main claims are supported by convincing evidence (in the form of adequate experimental evaluation, proofs, (pseudo-)code, references, assumptions).

**Q2-4 Reproducibility:**

2: Fair: key resources (e.g. proofs, code, data) are unavailable but key details (e.g. proof sketches, experimental setup) are sufficiently well-described for an expert to confidently reproduce the main results.

**Q3 Main Strengths:**

The proposed idea is well positioned within the state of the art and makes sense intuitively.
A good balance between theoretical and empirical support.
Clear evidence and claims.

**Q4 Main Weakness:**

A comparison to baseline time series approaches is lacking.

**Q5 Detailed Comments To The Authors:**

I would consider comparing to simpler time series classification baselines, particularly approaches for which UCR time series repository was initially collected.

**Q9 Complying With Reviewing Instructions:**

Yes

---

> ### Author Rebuttal · Authors · 2024-04-05
>
> We thank the reviewer for their constructive feedback and the recognition of our work's positioning within the current state of the art, its theoretical foundation, and the comprehensive experimental analysis.
>
> We acknowledge the suggestion to compare our approach with classical time series classification methods. It is important to note that our primary focus was developing a framework that generalizes randomized smoothing to arbitrary perturbations and incorporates methods for quantifying uncertainty, like conformal predictions and PAC. While we recognize that the ML model used may not represent the state-of-the-art architecture for time series analysis, both convolutional and transformer-based architectures offer versatility and are widely applied across numerous applications. Our framework is designed to work with any classifier; therefore, we do not expect different outcomes even for standard time series classifiers. In addition, the inclusion of standard time series classification approaches, while valuable, has been extensively covered in broader ML literature. Lastly, our decision to not make the in-house dataset publicly available at this time was due to ongoing validation of proprietary data. We appreciate this feedback and will consider the implications of extending our comparative analysis.
>
> We hope our responses address the reviewer concerns, and we look forward to any additional comments or feedback.

---

### Official Review · Reviewer_bgz8 · 2024-03-25

**Q2-1 Originality-Novelty:** 3
**Q2-2 Correctness-Technical Quality:** 3
**Q2-5 Clarity Of Writing:** 4

**Q1 Summary And Contributions:**

This paper introduces the Temporal Smooth Conformal Predictor (TSCP), a novel approach that extends Randomized Smoothing (RS) techniques to handle various transformations in time series data, motivated by the need to certify machine learning model robustness against domain-specific perturbations beyond adversarial attacks.
Key contributions include generalizing RS to handle transformations like time, window, and magnitude warping, introducing TSCP which utilizes temporal-shift transformations to smooth input data and demonstrate robustness against adversarial attacks and perturbations, and conducting experimental evaluations showcasing TSCP's effectiveness in robustness certification and empirical risk minimization on both an open-source time series dataset and a real-world in house sensor dataset.
Results indicate that TSCP maintains coverage and accuracy comparable to conventional conformal predictors, outperforming RSCP in certain configurations and is resilient to various transformations in the data domain.

**Q2-3 Extent To Which Claims Are Supported By Evidence:**

2: Fair: the main claims are somewhat supported by evidence (but the experimental evaluation may be weak, or does not match entirely with the claims, important baselines may be missing, proofs contain important ideas but lack rigor, algorithmic details are only discussed superficially, references are imprecise, assumptions are not sufficiently motivated or explicated, etc.).

**Q2-4 Reproducibility:**

4: Excellent: key resources (e.g. proofs, code, data) are available and key details (e.g. proof sketches, experimental setup) are comprehensively described for competent researchers to confidently and easily reproduce the main results.

**Q3 Main Strengths:**

The paper demonstrates originality by introducing TSCP, which extends randomized smoothing techniques to handle various transformations in time series data. This approach addresses a gap in existing research by focusing on domain-specific perturbations beyond adversarial attacks, particularly in the context of time series classification for automotive applications.

The integration of randomized smoothing with conformal prediction is technically sound, and the experimental evaluations are conducted with attention to detail, ensuring robustness and reliability of the results.

The writing in the paper is clear and well-structured. Complex technical concepts are explained in a manner that is understandable, and the organization of the paper aids in navigating through the development of TSCP and the presentation of experimental results.

**Q4 Main Weakness:**

While the TSCP approach introduces novel extensions to randomized smoothing for handling time series data, the paper could provide more explicit comparisons with existing literature or methods to highlight its novelty more effectively. Its effectiveness is my main doubt.

For the UCR time series dataset, RSCP has greater coverage than TSCP (98.2 vs 97.2), yet they have the 97.2 in bold. The TSCP may justify being chosen above the RSCP model since it provides a smaller set size (4.85 vs 4.79) which achieves the desired coverage, 0.9, but it is the RSCP model that should be in bold. I believe this is an incorrect interpretation of the results.
On the other hand, CP at 0.84 does not reach the target coverage 0.9, but the set size is a fraction of the others (1.09 vs 4.85 vs 4.79). I don't see the claims supported by the evidence.

In Figure 3, the differences between the approaches are not appreciable.  Overall, the new technique does not lead to a strong increase in the considered metrics.

**Q5 Detailed Comments To The Authors:**

1
There is an extra comma in "adversarial, attacks"

3.1
"after subtracting I_{tr}" should be D_{tr}

5.1
Briefly describe Chen dataset. You later call it "earth-quakes" datasets.


5.2

Table 1: according to your description, CP should be highlighted for coverage since 84 is closer to 90 than 97. However, to the best of my knowledge, a higher coverage is better. So, it would make sense to highlight the highest value and not the closer to 90%.

**Q9 Complying With Reviewing Instructions:**

Yes

---

> ### Author Rebuttal · Authors · 2024-04-05
>
> We thank the reviewer for their comprehensive and insightful feedback on our paper. Regarding the main strength of our work, we appreciate the acknowledgment of our novel TSCP. We understand the concerns raised about the explicit comparison with existing literature and the interpretation of our results, particularly regarding the coverage and set size metrics in relation to the RSCP model.
>
> To address the main weakness pointed out, the bold emphasis on TSCP over RSCP, despite the slight advantage of RSCP in coverage, was intended to highlight the practical significance of our approach in achieving a balance between coverage and set size, which we believe is critical for real-world applications. This approach aligns with findings from prior studies, such as Ghosh et al. [2023], illustrated in their Figure 3, where achieving the target coverage as close as possible is shown to enhance precision. Additionally, the choice for slightly exceeding the target coverage to ensure the inclusion of the true label within the prediction set is preferred for safety as the importance is on obtaining nominal levels of marginal coverage. Similarly in Figure 1 of Gendler et al. [2021], we can clearly perceive as the set size being very small compared to RSCP. However, the importance is on obtaining nominal levels of marginal coverage. Lastly, as demonstrated in Table 2 of our paper, TSCP can handle a wider range of possible transformations than RSCP.
>
> In response to detailed comments:
>
> - In our revisited version, we will provide a brief description of the Chen dataset to ensure clarity for the readers.
> - The interpretation of the table highlighting CP for coverage due to its closeness to the target 90% will be revisited. We aim to clarify that while higher coverage is generally sought, the efficiency and practicality of achieving target coverage with minimal set size are also critical considerations in our analysis.
>
> We thank the reviewer for their constructive critique and hope our responses address the reviewer concerns.

---

### Official Review · Reviewer_CRgf · 2024-03-26

**Q2-1 Originality-Novelty:** 3
**Q2-2 Correctness-Technical Quality:** 3
**Q2-5 Clarity Of Writing:** 3

**Q1 Summary And Contributions:**

This paper proposes a time series classification framework using conformalized random smoothing. The framework uses conformal predictions and randomised smoothing to achieve robustness against real-world pertubations in time series classification. The method seems to achieve good performance but I am not 100% convinced.

**Q2-3 Extent To Which Claims Are Supported By Evidence:**

3: Good: the main claims are supported by convincing evidence (in the form of adequate experimental evaluation, proofs, (pseudo-)code, references, assumptions).

**Q2-4 Reproducibility:**

2: Fair: key resources (e.g. proofs, code, data) are unavailable but key details (e.g. proof sketches, experimental setup) are sufficiently well-described for an expert to confidently reproduce the main results.

**Q3 Main Strengths:**

- The framework seems to work with any classifiers.
- This work is important for real world data with pertubations.

**Q4 Main Weakness:**

- There are lots of details about the framework but it is not exactly clear how everything fits in together. For example it is not clear how the conformal prediction set and PAC prediction set are produced.
- There are no comparison against state of the art methods.

**Q5 Detailed Comments To The Authors:**

This paper has a great potential and I look forward to see how it can be implemented in real life.

Although it seems like this framework can be applied to any soft classifiers, but it is not clear if it can be applied to non-deep learning methods.

There are lots of details about the framework but it is not exactly clear how everything fits in together. For example, it is not clear how the conformal prediction set and PAC prediction set are produced.

The adversarial experiment needs to be more specific. The authors should explain the Projected Gradient Descent that was used as the attack. How was it applied and whether it makes sense to apply to time series data.

Are there any specific reasons to use the CNN and transformer as the architecture choices for the datasets? How do the two models perform on the datasets? At the moment, the results seems bad compared to the state-of-the-art classifiers.

I am not sure how set size was calculated in the experiments.

I believe that state-of-the-art time series classification methods should also be compared to show that there is a strong need for this method.

In Table 1, CP has the smallest set-size, 4 times smaller than TSCP. It will be good to discuss this result.

In Table 5 of the appendix, the adversial accuracy seems to be averaged. Is that the correct way to aggregate this metric?

Why not use the new 128 UCR datasets instead of the old 85 datasets?

What are the distinct domain in the domain generalisation experiment? It is not clear what this experiment is about.

**Q9 Complying With Reviewing Instructions:**

Yes

---

> ### Author Rebuttal · Authors · 2024-04-05
>
> We thank the reviewer for their insightful feedback, which significantly contributes to improve the quality of our paper. We acknowledge the suggestions for clarity in framework integration, comparative analysis with state-of-the-art methods, and applicability to non-deep learning models.
>
> Here, we aim to respond to the feedback provided by the reviewer.
> - *Beyond Deep Learning:* It is important to highlight that our innovation is not in developing a new "optimal" time series classifier, but rather in introducing a methodology that can transform any classifier, regardless of whether it uses deep learning approaches or not, into a generalized smooth classifier. Initially, we focus on CNN and transformer-based network architectures due to their versatility across numerous data analysis contexts. However, the underlying approach of our framework is designed to be model-independent, making it applicable to any classifier that yields probabilistic outputs. This highlights its wide-ranging applicability.
> - *Framework Clarity:* In essence, the conformal prediction set leverages the model's probabilistic outputs to create prediction intervals with a specified confidence level. This approach allows for the construction of intervals that reflect the uncertainty of predictions, offering a quantifiable measure of confidence in the results. Meanwhile, the PAC prediction set refines these intervals to balance the size of the prediction set with the likelihood of correct coverage, even under distribution shifts. Our approach enhances uncertainty quantification through the integration of smooth classifiers with conformal prediction and PAC predictions, tailored for time series data. This dual approach strengthens the framework's ability to handle uncertainties across diverse domains, offering a comprehensive strategy for robust classification and uncertainty estimation in the face of data variability and potential adversarial challenges.
> - *Adversarial attack:* The Projected Gradient Descent (PGD) technique is a recognized approach for generating adversarial examples by altering inputs within a predefined norm boundary. This method essentially evaluates the most extreme modifications that can be made to an input, serving as a common benchmark for assessing the robustness of models in various studies. Despite its frequent use in benchmarks, the real-world applicability of PGD is somewhat restricted. Therefore, we have also chosen to explore perturbations in time series data, which are more aligned with the types of disturbances encountered in practical scenarios.
> - *Dataset Choice:* The choice to use the 85 UCR datasets over the newer 128 datasets was based on established benchmarks and comparability with prior research. However, we recognize the value of updating our analysis with the latest datasets to ensure relevance and comprehensiveness.
> - *Clarifications and Expansions:* Calculating the set size was relatively straightforward; similarly, to previous research, we simply measured the number of elements in the prediction set (cardinality of the set). With respect to the reduced set size highlighted in Table 1, we plan to address our framework's limitations and how these contribute to both the smaller and larger set sizes observed, drawing parallels to findings from [Gendler et al., 2021]. To ensure a fair comparison, we computed the average adversarial accuracy across 20 PGD attacks, which were evenly distributed across a range from 0 to 0.1. Although averaging over a range of attack strengths is not the conventional method for assessing adversarial accuracy, we believed it would offer a more comprehensive understanding of performance across various conditions, rather than focusing on outcomes at a single intensity level.
>
> In conclusion, we appreciate the constructive criticism and agree that addressing these points will significantly strengthen our paper. Our revisions will include a more detailed framework description, clarification on methodological choices, and additional insights into the framework's versatility.

---

### Meta-Review · Area_Chair_bbiz · 2024-04-22

This paper introduces the Temporal Smooth Conformal Predictor (TSCP). This is a novel framework that extends Randomized Smoothing (RS) techniques to handle various transformations in time series data, motivated by the need to certify machine learning model robustness against domain-specific perturbations beyond adversarial attacks. The framework uses conformal predictions and randomized smoothing to achieve robustness against real-world perturbations in time series classification.

I have carefully read the reviewers' comments and the responses, and I feel that the authors have adequately addressed all the major concerns. Hence, given the fact that the contributions are novel, I recommend acceptance.